# Compositional Modeling of Nonlinear Dynamical Systems with ODE-based Random Features

**Thomas M. McDonald**
Department of Computer Science
University of Sheffield
tmmcdonald1@sheffield.ac.uk

**Mauricio A. Álvarez**
Department of Computer Science
University of Sheffield
mauricio.alvarez@sheffield.ac.uk

## Abstract

Effectively modeling phenomena present in highly nonlinear dynamical systems whilst also accurately quantifying uncertainty is a challenging task, which often requires problem-specific techniques. We present a novel, domain-agnostic approach to tackling this problem, using compositions of physics-informed random features, derived from ordinary differential equations. The architecture of our model leverages recent advances in approximate inference for deep Gaussian processes, such as layer-wise weight-space approximations which allow us to incorporate random Fourier features, and stochastic variational inference for approximate Bayesian inference. We provide evidence that our model is capable of capturing highly nonlinear behaviour in real-world multivariate time series data. In addition, we find that our approach achieves comparable performance to a number of other probabilistic models on benchmark regression tasks.

## 1 Introduction

Dynamical systems are ubiquitous across the natural sciences, with many physical and biological processes being driven on a fundamental level by differential equations. Inferring ordinary differential equation (ODE) parameters using observational data from such systems is an active area of research [Meeds et al., 2019, Ghosh et al., 2021], however, in particularly complex systems it is often infeasible to characterise all of the individual processes present and the interactions between them. Rather than attempt to fully describe a complex system, latent force models (LFMs) [Alvarez et al., 2009] specify a simplified mechanistic model of the system which captures salient features of the dynamics present. This leads to a model which is able to readily extrapolate beyond the training input space, thereby retaining one of the key advantages of mechanistic modeling over purely data-driven techniques.

Modeling *nonlinear* dynamical systems presents an additional challenge, and whilst nonlinear differential equations have been incorporated into LFMs [Ward et al., 2020], shallow models such as LFMs are generally less capable than deep models of modeling the non-stationarities often present in nonlinear systems. Deep model architectures have greater representational power than shallow models as a result of their hierarchical structure, which typically consists of compositions of functions [LeCun et al., 2015]. Deep probabilistic models such as deep Gaussian processes (DGPs) [Damianou and Lawrence, 2013] are able to leverage this representational power, with the additional advantage of being able to accurately quantify uncertainty.

In this paper, we aim to model nonlinear dynamics by constructing a DGP from compositions of physics-informed random Fourier features, with the motivation behind this approach being that many real-world systems are compositional hierarchies [LeCun et al., 2015]. Whilst a number of recent works have incorporated random Fourier features into deep models [Cutajar et al., 2017, Mehrkanoon and Suykens, 2018], the only work we are aware of which does so in the context of a physics-informed DGP is that of Lorenzi and Filippone [2018]. The authors impose physical structure on a DGP with

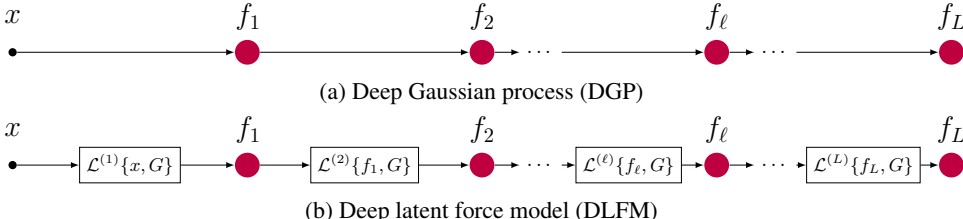

Figure 1: A conceptual explanation of how our proposed DLFM differs from a DGP. At each layer, we perform the operation $\mathcal{L}^{(\ell)}\{x, G\} = \int_0^x G^{(\ell)}(x - \tau)u(\tau)d\tau$, where $G$ is the Green's function corresponding to an ODE, and $u(\cdot)$ represents an exponentiated quadratic GP prior. For example, the second operation in the model shown above would take the form, $\mathcal{L}^{(2)}\{f_1, G\} = \int_0^{f_1} G^{(2)}(f_1 - \tau)u(\tau)d\tau$.

random features by constraining the function values within the model as a means of performing ODE parameter inference. Rather than following the approach of Lorenzi and Filippone [2018] and placing constraints on function values whilst using an exponentiated quadratic (EQ) kernel, we instead integrate a physics-informed prior into the structure of the DGP by utilising a kernel based on an ODE.

Our main contribution in this paper is the introduction of a novel approach to incorporating physical structure into a deep probabilistic model, whilst providing a sound quantification of uncertainty. We achieve this through derivation of physics-informed random Fourier features via the convolution of an EQ GP prior with the Green's function associated with a first order ODE. These features are then incorporated into each layer of a DGP, as shown in Figure 1. To ensure the scalability of our model to large datasets, stochastic variational inference is employed as a method for approximate Bayesian inference. We provide experimental evidence that our modeling framework is capable of capturing highly nonlinear dynamics effectively in both toy examples and real world data, whilst also being applicable to more general regression problems.

## 2   Background

In this section, we review the theory behind using random Fourier features for deep Gaussian processes. We also review the idea of building covariance functions for GPs that encode the dynamics of ODEs and how to compute those covariances using random Fourier features.

### 2.1   Deep Gaussian Processes with Random Feature Expansions

Gaussian processes (GPs) are non-parametric probabilistic models which offer a great degree of flexibility in terms of the data they can represent, however this flexibility is not unbounded. The hierarchical networks of GPs now widely known as deep Gaussian processes (DGPs) were first formalised by Damianou and Lawrence [2013], with the motivating factor behind their creation being the ability of deep networks to reuse features and allow for higher levels of abstraction in said features [Bengio et al., 2013], which results in such models having more representational power and flexibility than shallow models such as GPs. DGPs are effectively a composite function, where the input to the model is transformed to the output by being passed through a series of latent mappings (i.e. multivariate GPs). If we consider a supervised learning problem with inputs denoted by $\mathbf{X} = \{\mathbf{x}_n\}_{n=1}^N$ and targets denoted by $\mathbf{y} = \{y_n\}_{n=1}^N$, we can write the analytic form of the marginal likelihood for a DGP with $N_h$ hidden layers as $p(\mathbf{y}|\mathbf{X}, \theta) = \int p(\mathbf{y}|\mathbf{F}^{(N_h)})p(\mathbf{F}^{(N_h)}|\mathbf{F}^{(N_h-1)}, \theta^{(N_h-1)}) \dots p(\mathbf{F}^{(1)}|\mathbf{X}, \theta^{(0)})d\mathbf{F}^{(N_h)} \dots d\mathbf{F}^{(1)}$, where $\mathbf{F}^{(\ell)}$ and $\theta^{(\ell)}$ represent the latent values and covariance parameters respectively at the $\ell$-th layer, where $\ell = 0, \dots, N_h$. However, due to the need to propagate densities through non-linear GP covariance functions within the model, this integral is intractable [Damianou, 2015]. As this precludes us from employing exact Bayesian inference in these models, various techniques for approximate inference have been applied to DGPs in recent years, with most approaches broadly based upon either variational inference [Salimbeni and Deisenroth, 2017, Salimbeni et al., 2019, Yu et al., 2019] or Monte Carlo methods [Havasi et al., 2018].

Cutajar et al. [2017] outline an alternative approach to tackling this problem, which involves replacing the GPs present at each layer of the network with their two layer weight-space approximation, forming a Bayesian neural network which acts as an approximation to a DGP. If the $\ell$-th layer of such a DGP (consisting of zero-mean GPs with exponentiated quadratic kernels) receives an input $\mathbf{F}^{(\ell)}$ (where $\mathbf{F}^{(0)} = \mathbf{X}$), the random features for this layer are denoted by $\mathbf{\Phi}^{(\ell)} \in \mathbb{R}^{N \times 2N_{RF}^{(\ell)}}$ (where $N_{RF}^{(\ell)}$ denotes the number of random features used), and are given by,

$$\mathbf{\Phi}^{(\ell)} = \sqrt{\frac{(\sigma^2)^{(\ell)}}{N_{RF}^{(\ell)}}} \left[ \cos(\mathbf{F}^{(\ell)}\mathbf{\Omega}^{(\ell)}), \sin(\mathbf{F}^{(\ell)}\mathbf{\Omega}^{(\ell)}) \right], \tag{1}$$

where $(\sigma^2)^{(\ell)}$ is the marginal variance kernel hyperparameter, $N_{RF}^{(\ell)}$ is the number of random features used and $\mathbf{\Omega}^{(\ell)} \in \mathbb{R}^{D_{F^{(\ell)}} \times N_{RF}^{(\ell)}}$ is the matrix of spectral frequencies used to determine the random features. This matrix is assigned a prior $p(\Omega_d^{(\ell)}) = \mathcal{N}(\Omega_d^{(\ell)} \mid 0, (l^{(\ell)})^{-2})$ where $l^{(\ell)}$ is the lengthscale kernel hyperparameter, $D_{F^{(\ell)}}$ is the number of GPs within the layer, and $d = 1, ..., D_{F^{(\ell)}}$. The random features then undergo the linear transformation, $\mathbf{F}^{(\ell+1)} = \mathbf{\Phi}^{(\ell)}\mathbf{W}^{(\ell)}$, where $\mathbf{W}^{(\ell)} \in \mathbb{R}^{2N_{RF}^{(\ell)} \times D_{F^{(\ell+1)}}}$ is a weight matrix with each column assigned a standard normal prior. Training is achieved via *stochastic variational inference*, which involves establishing a tractable lower bound for the marginal likelihood and optimising said bound with respect to the mean and variance of the variational distributions over the weights and spectral frequencies across all layers of the network. The bound is also optimised with respect to the kernel hyperparameters across all layers.

## 2.2 Random Feature Approximations for Latent Force Models

In this paper, we take a similar approach to Cutajar et al. [2017], but we utilise physically-inspired random features. Physics-inspired covariance functions for GPs have been proposed by [Alvarez et al., 2009] under the name of latent force models, and more recently, the authors of Guarnizo and Álvarez [2018] studied the construction of random features related to such covariance functions.

Latent force models (LFMs) [Alvarez et al., 2009] are GPs which incorporate a physically-inspired kernel function, which typically encodes the behaviour described by a specific form of differential equation. Instead of taking a fully mechanistic approach and specifying all the interactions within a physical system, the kernel describes a simplified form of the system in which the behaviour is determined by $Q$ latent forces. Given an input data-point $t$ and a set of $D$ output variables $\{f_d(t)\}_{d=1}^D$, a LFM expresses each output as $f_d(t) = \sum_{q=1}^Q S_{d,q} \int_0^t G_d(t - \tau)u_q(\tau)d\tau$, where $G_d(\cdot)$ represents the Green's function for a certain form of linear differential equation, $u_q(t) \sim \mathcal{GP}(0, k_q(t, t'))$ represents the GP prior over the the $q$-th latent force, and $S_{d,q}$ is a sensitivity parameter weighting the influence of the $q$-th latent force on the $d$-th output. The general expression for the covariance of a LFM is given by $k_{f_d, f_{d'}}(t, t') = \sum_{q=1}^Q S_{d,q}S_{d',q} \int_0^t G_d(t - \tau) \int_0^{t'} G_{d'}(t' - \tau')k_q(\tau, \tau')d\tau'd\tau$.

Typically, an EQ form is assumed for the kernel governing the latent forces, $k_q(\cdot)$. Due to the linear nature of the convolution used to compute $f_d(t)$, the outputs can also be described by a GP. Exact inference in LFMs is tractable, however as with GPs, it scales with $\mathcal{O}(N^3)$ complexity [Rasmussen and Williams, 2006]. Furthermore, computing the double integration for $k_{f_d, f_{d'}}(t, t')$ leads to terms that include the numerical approximation of complex functions which burden the computation of the kernel function. By providing a random Fourier feature representation for the EQ kernel $k_q(t, t')$, Guarnizo and Álvarez [2018] were able to reduce this cubic dependence on the number of data-points to a linear dependence. This representation arises from Bochner's theorem, which states,

$$k_q(\tau, \tau') = e^{\frac{-(\tau - \tau')^2}{\ell_q^2}} = \int p(\omega)e^{j(\tau - \tau')\omega}d\omega \approx \frac{1}{N_{RF}} \sum_{s=1}^{N_{RF}} e^{j\omega_s \tau}e^{-j\omega_s \tau'}, \tag{2}$$

where $\ell_q$ is the lengthscale of the EQ kernel, $N_{RF}$ is the number of random features used in the approximation and $\omega_s \sim p(\omega) = \mathcal{N}(\omega | 0, 2/\ell_q^2)$. Substituting this form of the EQ kernel into $k_{f_d, f_{d'}}(t, t')$ leads to a fast approximation to the LFM covariance $k_{f_d, f_{d'}}(t, t') \approx \sum_{q=1}^Q \frac{S_{d,q}S_{d',q}}{N_{RF}} \left[ \sum_{s=1}^{N_{RF}} \phi_d(t, \theta_d, \omega_s)\phi_{d'}^*(t', \theta_{d'}, \omega_s) \right]$, where

$$\phi_d(t, \theta_d, \omega) = \int_0^t G_d(t - \tau)e^{j\omega\tau}d\tau, \tag{3}$$

with $\theta_d$ representing the Green's function parameters. When the Green's function is a real function, $\phi_{d'}^*(t', \theta_{d'}, \omega) = \phi_{d'}(t', \theta_{d'}, -\omega)$. Guarnizo and Álvarez [2018] refer to $\phi_d(t, \theta_d, \omega)$ as *random Fourier response features* (RFRFs).

## 3 Deep Latent Force Models

In this section, we present the deep latent force model (DLFM), a novel approach to incorporating physically-inspired prior beliefs into a deep Gaussian process. Firstly, we will outline the model architecture, before discussing our approach to training the model via stochastic variational inference.

### 3.1 Model Formulation

Rather than deriving the random features $\mathbf{\Phi}^{(\ell)}$ within the DGP from an EQ kernel [Cutajar et al., 2017], we instead populate this matrix with features derived from an LFM kernel. The exact form of the features derived is dependent on the Green's function used, which in turn depends on the form of differential equation whose characteristics we wish to encode within the model. Guarnizo and Álvarez [2018] derived a number of different forms corresponding to various differential equations, with the simplest case being that of a first order ordinary differential equation (ODE) of the form, $\frac{df(t)}{dt} + \gamma f(t) = \sum_{q=1}^{Q} S_q u_q(t)$, where $\gamma$ is a decay parameter associated with the ODE and $S_q$ is a sensitivity parameter which weights the effect of each latent force. For simplicity of exposition, we assume $D = 1$. The Green's function associated to this ODE has the form $G(x) = e^{-\gamma x}$. By using equation (3), the RFRFs associated with this ODE follow as

$$\phi(t, \gamma, \omega_s) = \frac{e^{j\omega_s t} - e^{-\gamma t}}{\gamma + j\omega_s}, \tag{4}$$

where compared to the expression (3), the parameter $\theta$ of the Green's function corresponds to the decay parameter $\gamma$. From here on, we redefine the spectral frequencies as $\omega_{q,s}$ to emphasise the fact that the values sampled are dependent on the lengthscale of the latent force by way of the prior, $\omega_{q,s} \sim \mathcal{N}(\omega|0, 2/\ell_q^2)$. We can collect all of the random features corresponding to the $q$-th latent force into a single vector, $\boldsymbol{\phi}_q^c(t, \gamma, \boldsymbol{\omega}_q) = \sqrt{S_q^2/N_{RF}}[\phi(t, \gamma, \omega_{q,1}), \cdots, \phi(t, \gamma, \omega_{q,N_{RF}})]^\top \in \mathbb{C}^{N_{RF} \times 1}$, where $\boldsymbol{\omega}_q = \{\omega_{q,s}\}_{s=1}^{N_{RF}}$, $\mathbb{C}$ refers to the complex plane and the super index $c$ in $\boldsymbol{\phi}_q^c(\cdot)$ makes explicit that this vector contains complex-valued numbers. By including the random features corresponding to all $Q$ latent forces within the model, we obtain $\boldsymbol{\phi}^c(t, \gamma, \boldsymbol{\omega}) = [(\boldsymbol{\phi}_1^c(t, \gamma, \boldsymbol{\omega}_1)^\top, \cdots, (\boldsymbol{\phi}_Q^c(t, \gamma, \boldsymbol{\omega}_Q))^\top]^\top \in \mathbb{C}^{QN_{RF} \times 1}$, with $\boldsymbol{\omega} = \{\boldsymbol{\omega}_q\}_{q=1}^{Q}$. These random features will be denoted as $\boldsymbol{\phi}_{LFM}^c(t, \gamma, \boldsymbol{\omega})$ to differentiate them from the features $\boldsymbol{\phi}(\cdot)$ computed from a generic EQ kernel.

**Higher-dimensional inputs** Although the expression for $\boldsymbol{\phi}_{LFM}^c(t, \gamma, \boldsymbol{\omega})$ was obtained in the context of an ODE where the input is the time variable, in this paper we exploit this formalism to propose the use of these features even in the context of a generic supervised learning problem where the input is a potentially high-dimensional vector $\mathbf{x} = [x_1, x_2, \cdots, x_p]^\top \in \mathbb{R}^{p \times 1}$. As will be noticed later, such an extension is also necessary if we attempt to use such features at intermediate layers of the composition. Essentially, we compute a vector $\boldsymbol{\phi}_{LFM}^c(x_m, \gamma_m, \boldsymbol{\omega}_m)$ for each input dimension $x_m$ leading to a set of vectors $\{\boldsymbol{\phi}_{LFM}^c(x_1, \gamma_1, \boldsymbol{\omega}_1), \cdots, \boldsymbol{\phi}_{LFM}^c(x_p, \gamma_p, \boldsymbol{\omega}_p)\}$. Notice that the samples $\boldsymbol{\omega}_m$ can also be different per input dimension, $x_m$. Although there are different ways in which these feature vectors can be combined, in this paper, we assume that the final random feature vector computed over the whole input vector $\mathbf{x}$ is given as $\boldsymbol{\phi}_{LFM}^c(\mathbf{x}, \boldsymbol{\gamma}, \boldsymbol{\Omega}) = \sum_{m=1}^{p} \boldsymbol{\phi}_{LFM}^c(x_m, \gamma_m, \boldsymbol{\omega}_m)$, where $\boldsymbol{\Omega} = \{\boldsymbol{\omega}_m\}_{m=1}^{p}$ and $\boldsymbol{\gamma} = \{\gamma_m\}_{m=1}^{p}$. An alternative to explore for future work involves expressing $\boldsymbol{\phi}_{LFM}^c(\mathbf{x}, \boldsymbol{\gamma}, \boldsymbol{\Omega})$ as $\boldsymbol{\phi}_{LFM}^c(\mathbf{x}, \boldsymbol{\gamma}, \boldsymbol{\Omega}) = \sum_{m=1}^{p} \alpha_m \boldsymbol{\phi}_{LFM}^c(x_m, \gamma_m, \boldsymbol{\omega}_m)$, with $\alpha_m \in \mathbb{R}$ a parameter that weights the contribution of each input feature differently. Although we allow each input dimension to have a different decay parameter $\gamma_m$ in the experiments in Section 5, for ease of notation we will assume that $\gamma_1 = \gamma_2 = \cdots = \gamma_p = \gamma$. For simplicity, we write $\boldsymbol{\phi}_{LFM}^c(\mathbf{x}, \gamma, \boldsymbol{\Omega})$. Therefore, $\boldsymbol{\phi}_{LFM}^c(\mathbf{x}, \gamma, \boldsymbol{\Omega})$ is a vector-valued function that maps from $\mathbb{R}^{p \times 1}$ to $\mathbb{C}^{QN_{RF} \times 1}$.

**Real version of the RFRFs** Rather than working with the complex-value random features $\boldsymbol{\phi}_{LFM}^c(\mathbf{x}, \gamma, \boldsymbol{\Omega})$, we can work with their real-valued counterpart by using $\boldsymbol{\phi}_{LFM}(\mathbf{x}, \gamma, \boldsymbol{\Omega}) =$

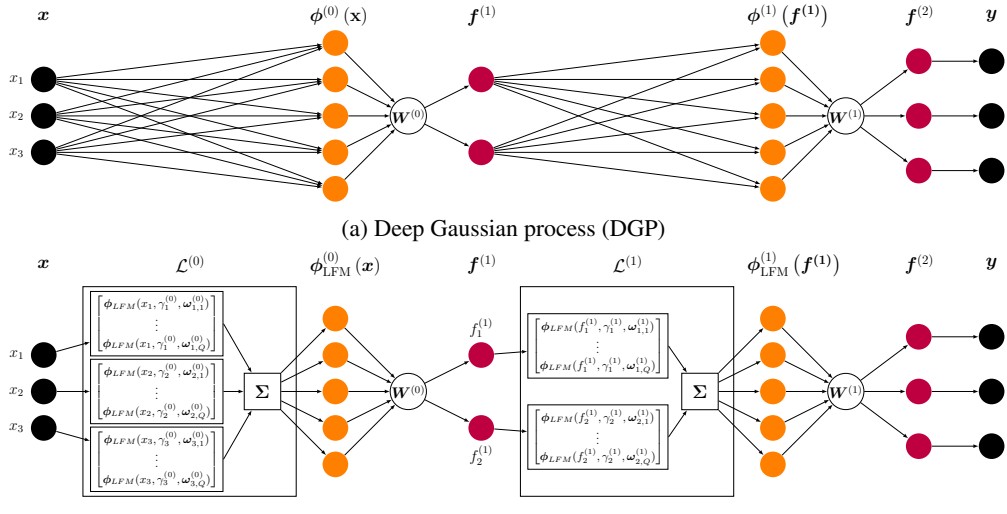

(a) Deep Gaussian process (DGP)

(b) Deep latent force model (DLFM)

Figure 2: An illustration of how our proposed model differs from a DGP with random feature expansions, with this example containing two layers. At each layer of the DLFM, for each input dimension, $N_{RF}$ random features of the form shown in (4) are computed for each of the $Q$ latent forces. The random feature vector $\phi_{LFM}^{(\ell)}$ is then formed by taking the sum of these features across the input dimensions. This summation is shown in the Figure by the block containing $\Sigma$.

$[(\mathfrak{Re}\{\phi_{LFM}^c(\mathbf{x},\gamma,\mathbf{\Omega})\})^\top, (\mathfrak{Im}\{\phi_{LFM}^c(\mathbf{x},\gamma,\mathbf{\Omega})\})^\top]^\top \in \mathbb{R}^{2QN_{RF}\times 1}$ Guarnizo and Álvarez [2018], where $\mathfrak{Re}(a)$ and $\mathfrak{Im}(a)$ take the real component and imaginary component of $a$, respectively. For an input matrix $\mathbf{X} = [\mathbf{x}_1,\cdots,\mathbf{x}_N]^\top \in \mathbb{R}^{N\times P}$, the matrix $\mathbf{\Phi}_{LFM}(\mathbf{X},\gamma,\mathbf{\Omega}) = [\phi_{LFM}(\mathbf{x}_1,\gamma,\mathbf{\Omega}),\cdots,\phi_{LFM}(\mathbf{x}_N,\gamma,\mathbf{\Omega})]^\top \in \mathbb{R}^{N\times 2QN_{RF}}$.

**Composition of RFRFs** We now use $\mathbf{\Phi}_{LFM}(\mathbf{X},\gamma,\mathbf{\Omega})$ as a building block of a layered architecture of RFRFs. We write $\mathbf{\Phi}_{LFM}^{(\ell)}(\mathbf{F}^{(\ell)},\gamma^{(\ell)},\mathbf{\Omega}^{(\ell)}) \in \mathbb{R}^{N\times 2Q^{(\ell)}N_{RF}^{(\ell)}}$ and follow a similar construction to the one described in Section 2.1, where $\mathbf{F}^{(\ell+1)} = \mathbf{\Phi}_{LFM}^{(\ell)}(\mathbf{F}^{(\ell)},\gamma^{(\ell)},\mathbf{\Omega}^{(\ell)})\mathbf{W}^{(\ell)}$, and $\mathbf{W}^{(\ell)} \in \mathbb{R}^{2Q^{(\ell)}N_{RF}^{(\ell)}\times D_{F^{(\ell+1)}}}$. As before, $\mathbf{F}^{(0)} = \mathbf{X}$. Figure 2 is an example of this compositional architecture of RFRFs, which we refer to as a *deep latent force model* (DLFM). When considering multiple-output problems, we allow extra flexibility to the decay parameters and lengthscales at the final ($L$-th) layer such that they vary not only by input dimension, but also by output. Mathematically, this corresponds to computing $\mathbf{F}_d^{(L)} = \mathbf{\Phi}_{LFM}^{(L-1)}(\mathbf{F}^{(\ell)},\gamma_d^{(L-1)},\mathbf{\Omega}_d^{(L-1)})\mathbf{W}_d^{(L-1)}$, where $d = 1,...,D$.

**Model interpretation** In a DGP, the layers of GPs successively warp the input space, propagating uncertainty through the model. The DLFM also performs a similar warping of the inputs, however, in addition, the Green's function involved in the convolution which takes place within our model can be interpreted as a filter, which selectively filters out some of the frequencies of the GPs at each layer. Therefore, the DLFM is not only warping the intermediate inputs, but also filtering components in the latent GPs, altering their degree of smoothness.

### 3.2 Variational Inference

As previously mentioned, exact Bayesian inference is intractable for models such as ours, therefore in order to train the DLFM we employ stochastic variational inference [Hoffman et al., 2013]. For notational simplicity, let $\mathbf{W}$, $\mathbf{\Omega}$ and $\mathbf{\Theta}$ represent the collections of weight matrices, spectral frequencies and kernel hyperparameters respectively, across all layers of the model. We seek to optimise the variational distributions over $\mathbf{\Omega}$ and $\mathbf{W}$ whilst also optimising $\mathbf{\Theta}$, however we do not place a variational distribution over these hyperparameters. Our approach resembles the VAR-FIXED training strategy described by Cutajar et al. [2017] which involves reparameterizing $\Omega_{ij}^{(\ell)}$ such that $\Omega_{ij}^{(\ell)} = s_{ij}^{(\ell)}\epsilon_{rij}^{(\ell)} + m_{ij}^{(\ell)}$, where $m_{ij}^{(\ell)}$ and $(s^2)_{ij}^{(\ell)}$ represent the means and variances associated with

the variational distribution over $\Omega_{ij}^{(\ell)}$, and ensuring that the standard normal samples $\epsilon_{rij}^{(\ell)}$ are fixed throughout training rather than being resampled at each iteration. If we denote $\boldsymbol{\Psi} = \{\mathbf{W}, \boldsymbol{\Omega}\}$ and consider training inputs $\mathbf{X} \in \mathbb{R}^{N \times D_{in}}$ and outputs $\mathbf{y} \in \mathbb{R}^{N \times D_{out}}$, we can derive a tractable lower bound on the marginal likelihood using Jensen's inequality, which allows for minibatch training using a subset of $M$ observations from the training set of $N$ total observations. This bound, derived by Cutajar et al. [2017], takes the form,

$$\log[p(\mathbf{y}|\mathbf{X}, \boldsymbol{\Theta}] = E_{q(\boldsymbol{\Psi})} \log[p(\mathbf{y}|\mathbf{X}, \boldsymbol{\Psi}, \boldsymbol{\Theta})] - \mathrm{DKL}[q(\boldsymbol{\Psi})||p(\boldsymbol{\Psi}|\boldsymbol{\Theta})] \tag{5}$$

$$\approx \left[ \frac{N}{M} \sum_{k \in \mathcal{I}_M} \frac{1}{N_{\mathrm{MC}}} \sum_{r=1}^{N_{\mathrm{MC}}} \log[p(\mathbf{y}_k|\mathbf{x}_k, \tilde{\boldsymbol{\Psi}}_r, \boldsymbol{\Theta})] \right] - \mathrm{D_{KL}}[q(\boldsymbol{\Psi})||p(\boldsymbol{\Psi}|\boldsymbol{\Theta})], \tag{6}$$

where $\mathrm{D_{KL}}$ denotes the KL divergence, $\tilde{\boldsymbol{\Psi}}_r \sim q(\boldsymbol{\Psi})$, the minibatch input space is denoted by $\mathcal{I}_M$ and $N_{\mathrm{MC}}$ is the number of Monte Carlo samples used to estimate $E_{q(\boldsymbol{\Psi})}$. $q(\boldsymbol{\Psi})$ and $p(\boldsymbol{\Psi}|\boldsymbol{\Theta})$ denote the approximate variational distribution and the prior distribution over the parameters respectively, both of which are assumed to be Gaussian in nature. A full derivation of this bound and the expression for the KL divergence between two normal distributions are included in the supplemental material.

We mirror the approach of Cutajar et al. [2017] and Kingma and Welling [2014] by reparameterizing the weights and spectral frequencies, which allows for stochastic optimisation of the means and variances of our distributions over $\boldsymbol{W}$ and $\boldsymbol{\Omega}$ via gradient descent techniques. Specifically, we use the AdamW optimizer [Loshchilov and Hutter, 2018], implemented in PyTorch, as empirically it led to superior performance compared to other alternatives such as conventional stochastic gradient descent.

## 4  Related Work

As mentioned previously, the work of Lorenzi and Filippone [2018] also aims to incorporate physical structure into a deep probabilistic model, however the authors achieve this by constraining the dynamics of their DGP with random feature expansions, rather than specifying a physics-informed kernel as in our approach. Additionally, the model developed by Lorenzi and Filippone [2018] is primarily designed for tackling the problem of ODE parameter inference in situations where the form of the underlying ODEs driving the behaviour of a system are known. In contrast, our DLFM does not assume any knowledge of the differential equations governing the complex dynamical systems we study in Section 5, and thus we do not attempt to perform parameter inference; our primary aim is to construct a robust, physics-inspired model with extrapolation capabilities and the ability to quantify uncertainty in its predictions.

The work of Mehrkanoon and Suykens [2018] is another example of a deep model in which each layer consists of a feature mapping computed using random Fourier features, followed by a linear transformation. Whilst this bears a similarity to the architecture of our model and to that of Lorenzi and Filippone [2018], the authors only consider an EQ kernel for their feature mapping, with no consideration given to physically-inspired features or constraining function dynamics. Wang et al. [2019] also propose a scalable deep probabilistic model capable of modeling non-linear functions (they specifically consider multivariate time series), but their approach relies on a deep neural network to model global behaviour, whilst relying on GPs at each layer to capture random effects and local variations specific to individual time series. Zammit-Mangion et al. [2021] is another example of recent work which aims to model nonstationary processes using a deep probabilistic architecture, with their primary focus being spatial warping. In Duncker and Sahani [2018], the authors propose a nested GP model in which the inner GP warps the temporal input it is provided. If we consider a univariate time input, our model performs a similar process, with the weight-space GP layer warping the temporal input and passing the output into a second GP layer. However, unlike in Duncker and Sahani [2018], our framework allows for an arbitrary number of warping transformations.

We wish to clarify that although our model is based on LFMs defined by convolutions of the form used to compute $f_d(t)$ in Section 2.2, our model differs from the convolutional DGP construction outlined by Dunlop et al. [2018], $f^{(\ell+1)}(t) = \int f^{(\ell)}(t - \tau) u^{(\ell+1)}(\tau) d\tau$, which the authors argue results in trivial behaviour with increasing depth. Instead, our model takes the form, $f^{(\ell+1)}(f^{(\ell)}) = \int_0^{f^{(\ell)}} G(f^{(\ell)} - \tau^{(\ell+1)}) u^{(\ell+1)}(\tau^{(\ell+1)}) d\tau^{(\ell+1)}$, which is more akin to the compositional construction outlined by the Dunlop et al. [2018], just with a kernel which happens to involve a convolution.

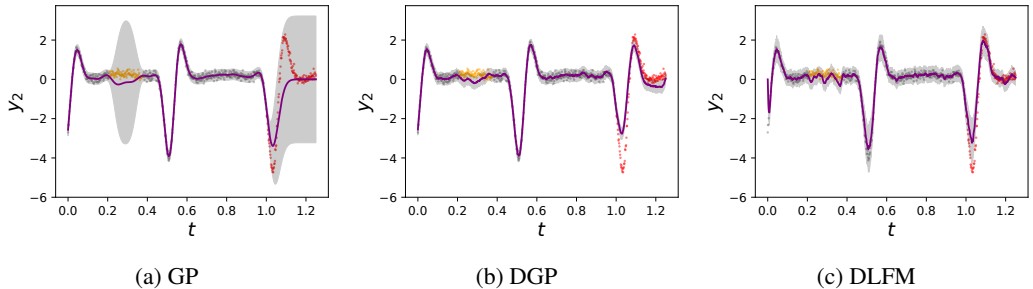

| (a) GP | (b) DGP | (c) DLFM |

Figure 3: Three model fits to data from our toy dynamical system with the latent function $u(t) = \cos(0.5t) + 6\sin(3t)$. Grey, orange and red data-points represent training data, interpolation test data and extrapolation test data respectively. The purple curves represent the predictive means, and the shaded areas represent $\pm 2\sigma$.

## 5 Experiments

All of the experimental results in this section were obtained using a single node of a cluster, consisting of a 40 core Intel Xeon Gold 6138 CPU and a NVIDIA Tesla V100 SXM2 GPU with 32GB of RAM. Unless otherwise specified, all models in this section were implemented in pure PyTorch, trained using the AdamW optimizer with a learning rate of 0.01 and a batch size of 1000.[1] The DLFMs and DGPs with random feature expansions [Cutajar et al., 2017] tested all utilised a single hidden layer of dimensionality $D_{F^{(\ell)}} = 3$, 100 Monte Carlo samples and 100 random features per layer. Further details regarding experimental setups are provided in the supplemental material, alongside additional experimental results.

### 5.1 Compositional System Toy Problem

Firstly, to verify that our model is capable of modeling the compositional systems which its architecture resembles, we train the model on the vector-valued inputs $t$ and noise-corrupted outputs $y_2(t) = f_2(t) + \epsilon$ of a system characterised by $f_1(t) = \int_0^t G_1(t - \tau)u(\tau)d\tau$ and $f_2(f_1) = \int_0^{f_1} G_2(f_1 - \tau')u(\tau')d\tau'$, where $G_1(\cdot)$ and $G_2(\cdot)$ represent the Green's functions corresponding to the first order ODEs which these two integrals represent the solutions to, and $\epsilon \sim \mathcal{N}(0, 0.04)$. We evaluate an exact GP, a DLFM (with a single latent force and 100 random features) and a DGP with random features, with a batch size equal to the size of the training set. The latent function is sinusoidal in nature, resulting in a signal consisting of pulses with varying amplitude. From the model predictions shown in Figure 3, we see that all three models capture the behaviour of the system well in the training regime, but only the deep models are able to extrapolate. The DLFM and DGP have similar normalised mean squared errors (NMSE) on the extrapolation test set of 1.5 and 1.4 respectively, however, the DLFM appears to provide a more realistic quantification of predictive uncertainty, which is supported by the fact that the mean negative log likelihood (MNLL) evaluated on the extrapolation test set for the DLFM was 2.4, whereas for the DGP it was 3.7.

### 5.2 PhysioNet Multivariate Time Series

To assess the ability of our model to capture highly nonlinear behaviour, we evaluate its performance on a subset of the CHARIS dataset (ODC-BY 1.0 License) [Kim et al., 2016], which can be found on the PhysioNet data repository [Goldberger et al., 2000]. The data available for each patient consists of an electrocardiogram (ECG), alongside arterial blood pressure (ABP) and intracranial pressure (ICP) measurements; all three of these signals are sampled at regular intervals and are nonlinear in nature. We use a subset of this data consisting of the first 1000 time steps for a single patient. We test two variations of the DLFM in this section, one with two latent forces and 50 random features per force ($Q = 2$), and one with a single latent force and 100 random features ($Q = 1$). To clarify, both of these models contain 100 random features per layer, but in the $Q = 2$ case, 50 features are

---

[1]Our code is publicly available in the repository: `https://github.com/tomcdonald/Deep-LFM`. The code was also included within the supplemental material at the time of review.

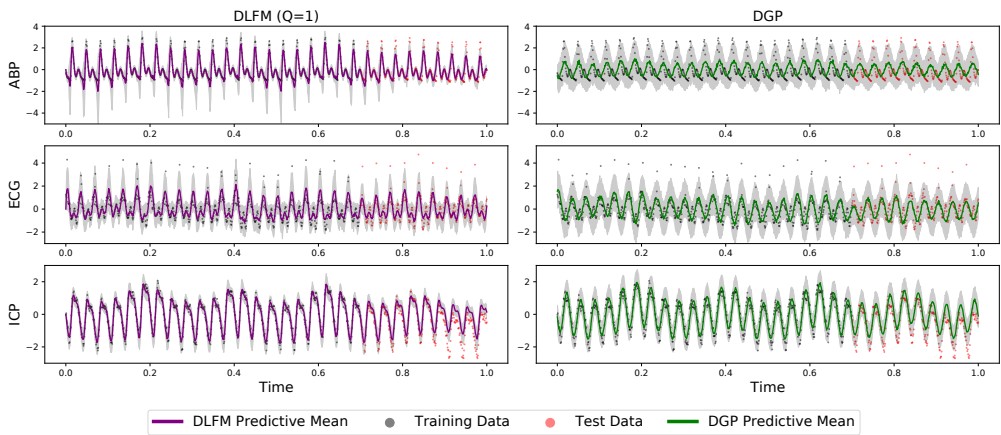

Figure 4: Predictions generated for each of the three outputs within the CHARIS dataset from a DLFM with one latent force (in the left-hand column), and a DGP with an exponentiated quadratic kernel (in the right-hand column). The shaded grey areas in each plot represent $\pm 2\sigma$.

Table 1: Extrapolation test set results for each output within the CHARIS dataset, with standard error in brackets.

| | ABP | | ECG | | ICP | |
|---|---|---|---|---|---|---|
| | NMSE | MNLL | NMSE | MNLL | NMSE | MNLL |
| DLFM ($Q = 1$) | **0.39 (0.05)** | **0.75 (0.07)** | **0.45 (0.04)** | **0.99 (0.05)** | 0.36 (0.04) | 0.90 (0.06) |
| DLFM ($Q = 2$) | 0.42 (0.07) | 0.77 (0.08) | 1.58 (0.98) | 1.16 (0.16) | **0.32 (0.06)** | **0.88 (0.11)** |
| DGP-EQ | 0.73 (0.002) | 1.21 (0.001) | 0.66 (0.01) | 1.22 (0.01) | 0.44 (0.01) | 1.00 (0.01) |
| DGP-ARC | 1.01 (0.001) | 2.60 (0.02) | 1.03 (0.001) | 2.73 (0.04) | 1.36 (0.003) | 3.25 (0.02) |
| VAR-GP | 1.01 (0.0002) | 26.90 (9.02) | 1.02 (0.001) | 4.98 (0.10) | 1.33 (0.003) | 15.50 (6.48) |
| DNN | 1.04 (0.01) | N/A | 1.08 (0.02) | N/A | 2.41 (0.09) | N/A |
| LFM-RFF | 1.00 (0.003) | N/A | 1.11 (0.02) | N/A | 1.02 (0.01) | N/A |

derived from two distinct latent forces. We compare these models to two DGPs with random features, employing EQ (DGP-EQ) and arccosine (DGP-ARC) kernels, as well as a deep neural network (DNN) with a single hidden layer of dimensionality 300, ReLU activations, and a dropout rate of 0.5. In addition to deep models, we also evaluate the performance of a heterogeneous multi-output GP trained using stochastic variational inference (VAR-GP) [Moreno-Muñoz et al., 2019] and a shallow LFM with random Fourier features derived from the same first order ODE as our model (LFM-RFF) [Guarnizo and Álvarez, 2018], consisting of two latent forces and 20 random features. Experimental results comparing the ability of each of these models to impute values within the training-input space are presented in the supplemental material, however we focus here on the more challenging task of extrapolating beyond the training input-space by training the aforementioned models on the first 700 observations and withholding the remaining 300 as a test set.

The results in Table 1 show that our model outperforms the other techniques tested, with the single latent force variant in particular converging to a significantly lower NMSE and MNLL than the other approaches across all three outputs. From these results we can conclude that the improved performance we see is a result of the ODE-based random features in the DLFM allowing the model to extrapolate accurately beyond the training input-space, rather than purely due to the additional flexibility afforded by allowing an increased number of lengthscales per layer, as the $Q = 1$ model contains the same number of lengthscales per layer as the DGP. We also find that the deep models tested consistently outperform the shallow models. From Figure 4, we can see that the DLFM is more able to accurately extrapolate the nonlinear dynamics present in the system than the DGP, especially the variations in ABP.

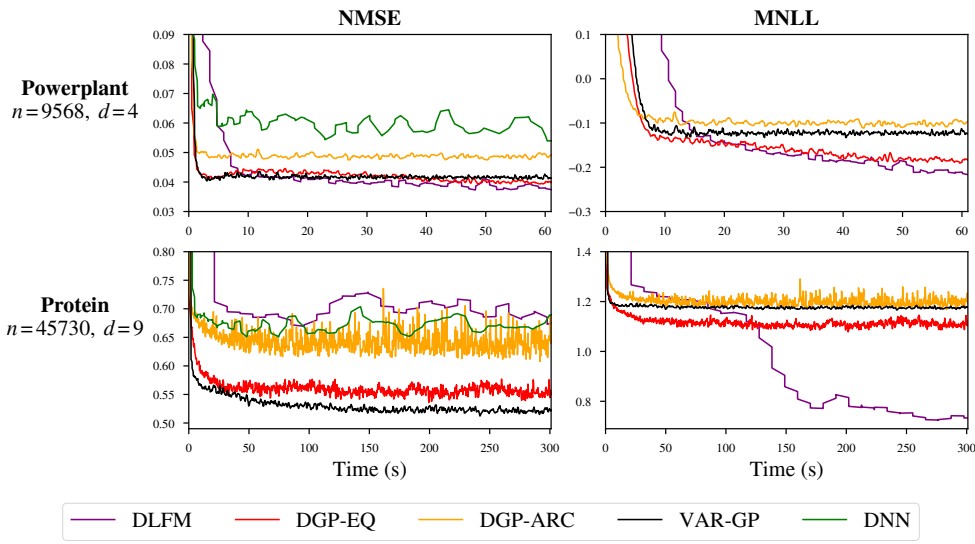

Figure 5: Progression of validation set metrics on the UCI benchmarks, averaged over three folds.

Table 2: Test set results on the UCI benchmarks, with standard error in brackets.

|  | Powerplant | | Protein | |
|---|---|---|---|---|
|  | NMSE | MNLL | NMSE | MNLL |
| DLFM | **0.0557 (0.0005)** | **- 0.017 (0.005)** | 0.698 (0.004) | **0.754 (0.006)** |
| DGP-EQ | 0.0581 (0.0002) | 0.016 (0.003) | 0.553 (0.005) | 1.101 (0.002) |
| DGP-ARC | 0.0619 (0.0002) | 0.039 (0.002) | 0.644 (0.027) | 1.203 (0.008) |
| VAR-GP | 0.0576 (0.0001) | 0.050 (0.001) | **0.520 (0.005)** | 1.174 (0.002) |
| DNN | 0.0754 (0.0011) | N/A | 0.664 (0.010) | N/A |

### 5.3 UCI Regression Benchmarks

Finally, we also evaluated the performance of the model on two regression datasets from the UCI Machine Learning Repository [Dua and Graff, 2017], 'Powerplant' and 'Protein', which both consist of a multivariate input and a univariate target. For both of these experiments, we evaluate the same DLFM, DGP and DNN architectures as for the PhysioNet experiment, alongside a multi-task variational GP (VAR-GP) [Hensman et al., 2015] implemented using GPyTorch [Gardner et al., 2018]. Achieving high performance on benchmark regression datasets was not the primary motivation behind this work, however from the results shown in Figure 5 and Table 2, we find that the DLFM exhibits comparable performance to the models described above. Most notably, our model consistently converges to a superior mean negative log likelihood (MNLL), both on the validation set during training and also on the held-out test set.

## 6 Conclusion

We have presented a novel approach to modeling highly nonlinear dynamics with a sound quantification of uncertainty, using compositions of random features derived from a first order ordinary differential equation (ODE). Our results show that our model is able to effectively capture nonlinear dynamics in both single and multiple output settings, with the added benefit of competitive performance on benchmark regression datasets.

Whilst we did find that utilising a kernel based on a first order ODE yielded superior results compared to competing models, the simplicity of the ODE on which our model is based limits its flexibility. However, random features can also be derived from more expressive kernels based on higher order ODEs [Guarnizo and Álvarez, 2018] and even partial differential equations. Integrating such kernels

into our modeling framework, and perhaps even varying the kernel choice between layers, would be interesting avenues for future research. Another limitation of our approach is that whilst we give $W$ and $\Omega$ a variational treatment, we do not extend this to the kernel hyperparameters $\Theta$; this could also be considered in future work.

In this work, we have demonstrated the applicability of our model to healthcare data in Section 5.2. Whilst our results demonstrate the capability of our model to learn complex dynamics within a highly nonlinear system, it is important to note that relying heavily on machine learning (ML) software within the domain of healthcare presents the possibility for negative societal impact, specifically ethical and practical issues. Care must be taken to ensure that such software is thoroughly tested and that appropriate safeguards are in place to avoid situations which could cause possible harm to patients. For example, clinical decisions should be made with input from a trained professional, rather than purely based upon information from an ML system.

## Acknowledgments and Disclosure of Funding

Thomas M. McDonald thanks the Department of Computer Science at the University of Sheffield for financial support. Mauricio A. Álvarez has been financed by the EPSRC Research Projects EP/R034303/1, EP/T00343X/2 and EP/V029045/1.

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
