# A  Appendix

Within this appendix, we firstly present a table-based comparison of our proposed model with a number of relevant approaches mentioned in the main text. Following this, additional details regarding experimental setups and further justification for our experimental design choices are supplied. We also provide a number of additional experimental results; firstly, we consider an imputation problem using the PhysioNet data from the main paper, before also evaluating deeper variations of the models from the main paper on the PhysioNet extrapolation task and the UCI benchmark datasets. In addition, we evaluate our model on the *FitzHugh-Nagumo* ODE system, the chaotic *Lorenz attractor* system, provide an experimental comparison between our model and *neural processes*, and also include a brief hyperparameter study. We also present a detailed derivation of our variational lower bound, alongside the expression for the Kullback-Leibler divergence between two Gaussian distributions. Finally, following a short discussion on computational complexity, we include the NeurIPS 2021 author checklist.

## A.1  Model Comparison

Presented in Table A1 is an overview of a number of probabilistic modeling techniques which are relevant to the work we present in this paper.

## A.2  Experimental Setups

For all of the experiments performed, we concatenate the output of each layer within the DLFM with the original input to the model as a means of avoiding pathological behaviour [Duvenaud et al., 2014]. We also extend this treatment to all of the DGPs tested. The lengthscales and decay parameters within the DLFM are all initialised to 0.01, except at the output layer where we initialise the lengthscales to 1.0. The likelihood variance is also initialised to 0.01, whilst the sensitivity parameters are randomly initialised from a standard normal distribution. We attempted to closely mirror this setup for the DGPs tested by initialising the lengthscales to 0.01, and whilst this lead to improved performance on the dynamical system experiments, such an initialisation resulted in poor performance on the UCI regression benchmarks, therefore we reverted to the initialisation used by the original implementation ($log(D_{F^{(\ell)}})$, where $D_{F^{(\ell)}}$ denotes the dimensionality of the $\ell$-th layer).

**Toy Data Experiment**  The data was generated by solving the hierarchical ODE system with $\gamma_1 = 0.01$, $\gamma_2 = 0.02$, $\omega = 1$, $\tau = 0$ and the initial values of $f_1$ and $f_2$ set equal to zero. The extrapolation test data consists of 150 evenly spaced data-points between $t = 1.0$ and $t = 1.25$, whilst the interpolation test data consists of 100 evenly spaced data-points between approximately $t = 0.208$ and $t = 0.375$.

**PhysioNet Experiments**  For the extrapolation experiment, the test data range consisted of the 300 data-points lying between $t = 0.7$ and $t = 1.0$. For the imputation experiment included in the supplemental material, the test data consisted of 150 contiguous, non-overlapping data-points from each of the three outputs; these were in the range $t = 0.20$ to $t = 0.35$ for ABP, $t = 0.40$ to $t = 0.55$ for ECG and $t = 0.60$ to $t = 0.75$ for ICP.

**UCI Experiments**  We utilised the same train and test folds as Cutajar et al. [2017] in order to make our results as directly comparable as possible.[1] From each training fold, we set aside 1% of the observations as a validation set.

## A.3  Additional Experiments

### A.3.1  PhysioNet Imputation

In this experiment, we test the ability of each model to impute missing output values, by removing 150 non-overlapping data-points from each output during training and using these data-points as a test set. From the results shown in Table A2 and Figure A1, we find that although the DGP is capable of

---

[1]The UCI train and test folds are available at `https://github.com/mauriziofilippone/deep_gp_random_features/tree/master/code/FOLDS`.

Table A1: A high-level comparison of the DLFM with a number of other approaches. This includes whether each of the probabilistic models listed is deep or shallow, and whether they incorporate any physical dynamics, followed by a brief summary of the approach.

| Model | Deep? | Physics-Informed? | Summary |
|---|---|---|---|
| LFM [Alvarez et al., 2009] | × | ✓ | GP kernel derived from ODEs |
| LFM-RFF [Guarnizo and Álvarez, 2018] | × | ✓ | Random Fourier features derived from ODEs |
| DLFM (our work) | ✓ | ✓ | Random Fourier features derived from ODEs integrated into DGP architecture |
| C-DGP [Lorenzi and Filippone, 2018] | ✓ | ✓ | ODE-based constraints applied to dynamics of a DGP with random Fourier features |
| DGP-RFF [Cutajar et al., 2017] | ✓ | × | Random Fourier feature expansions applied to a DGP architecture |
| Deep Hybrid Neural Kernel [Mehrkanoon and Suykens, 2018] | ✓ | × | Bridges deep learning and kernel methods using random Fourier features |
| Deep Factors [Wang et al., 2019] | ✓ | × | Probabilistic approach to modeling nonlinear systems which employs GPs and DNNs |
| IWGP & SDSP [Zammit-Mangion et al., 2021] | ✓ | × | Modeling of non-stationary data with compositions of stochastic processes |
| tw-pp-svGPFA [Duncker and Sahani, 2018] | ✓ | × | Nested GP-based model which infers latent time warping functions |
| Conv-CNP [Gordon et al., 2019] | ✓ | × | An extension of a neural process which is capable of modeling translation equivariance |
| GNP [Bruinsma et al., 2021] | ✓ | × | A variant of the Conv-CNP which also models correlations |

imputing the missing outputs with some success, the DLFMs converge to a lower NMSE and MNLL across all three of the outputs. The MNLL metrics for the LFM-RFF are omitted from Table A2 as the model returned negative predictive variances.

### A.3.2 Additional Hidden Layers

We carried out two additional experiments to assess the performance of our DLFM with two hidden layers, rather than one. For comparison, we also evaluated two hidden layer variations of the DGP-EQ, DGP-ARC and DNN models mentioned in the main body of the paper. All other aspects of these four models were kept identical to the experimental setups described in the main text.

**PhysioNet Multivariate Time Series** Considering once again the extrapolation problem described in the main paper, Table A3 contains the test set NMSE and MNLL corresponding to each output, for each of the models mentioned above, with associated standard error reported with respect to the random seed. The results from evaluating the DLFMs are largely similar to the single hidden layer case, with the $Q = 1$ variant continuing to outperform all of the other techniques tested. We found

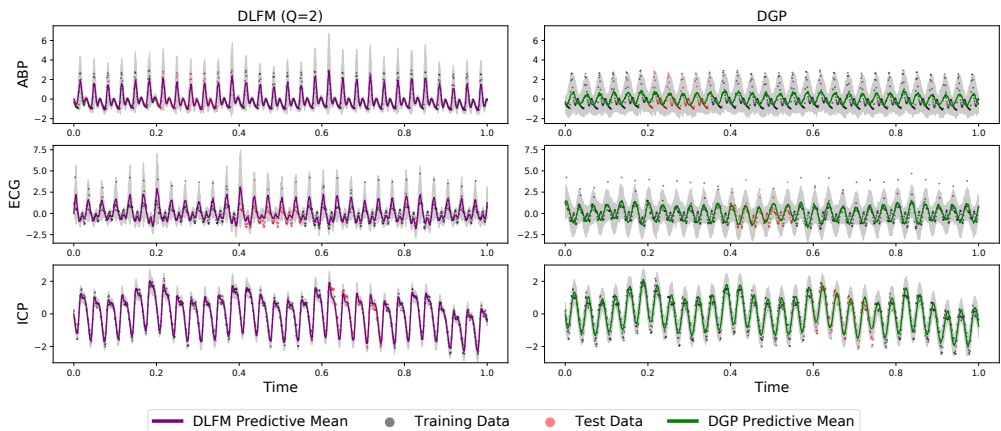

Figure A1: Predictions generated from the imputation scenario described in Section A.3.1 for each of the three outputs within the CHARIS dataset from a DLFM with two latent forces (in the left-hand column), and a DGP with an exponentiated quadratic kernel (in the right-hand column). The shaded grey areas in each plot represent $\pm 2\sigma$.

Table A2: Imputation test set results for each output of the CHARIS dataset, with the standard error reported in brackets.

| | ABP | | ECG | | ICP | |
|---|---|---|---|---|---|---|
| | NMSE | MNLL | NMSE | MNLL | NMSE | MNLL |
| DLFM ($Q = 1$) | 0.42 (0.06) | 0.79 (0.05) | **0.48 (0.04)** | 1.00 (0.05) | 0.19 (0.02) | 0.60 (0.04) |
| DLFM ($Q = 2$) | **0.27 (0.11)** | **0.49 (0.11)** | 0.68 (0.13) | **1.05 (0.04)** | **0.13 (0.01)** | **0.46 (0.02)** |
| DGP-EQ | 0.73 (0.02) | 1.18 (0.01) | 0.63 (0.02) | 1.21 (0.01) | 0.23 (0.02) | 0.73 (0.04) |
| DGP-ARC | 1.00 (0.0003) | 8.10 (0.03) | 1.02 (0.001) | 5.00 (0.02) | 1.03 (0.0004) | 3.60 (0.02) |
| VAR-GP | 1.00 (0.0001) | 23.9 (13.1) | 1.03 (0.0002) | 10.3 (1.3) | 1.00 (0.0002) | 19.2 (4.1) |
| DNN | 1.00 (0.001) | N/A | 1.01 (0.001) | N/A | 0.99 (0.001) | N/A |
| LFM-RFF | 1.00 (0.001) | N/A | 1.01 (0.01) | N/A | 0.97 (0.02) | N/A |

Table A3: Extrapolation test set results for the two hidden layer models, for each output within the CHARIS dataset, with standard error in brackets.

| | ABP | | ECG | | ICP | |
|---|---|---|---|---|---|---|
| | NMSE | MNLL | NMSE | MNLL | NMSE | MNLL |
| DLFM ($Q = 1$) | **0.26 (0.08)** | **0.45 (0.13)** | **0.55 (0.09)** | **1.03 (0.08)** | **0.35 (0.07)** | **0.92 (0.15)** |
| DLFM ($Q = 2$) | 0.72 (0.21) | 1.79 (0.57) | 0.91 (0.21) | 1.37 (0.19) | 0.54 (0.14) | 1.45 (0.33) |
| DGP-EQ | 1.01 (0.0001) | 27.1 (1.13) | 1.00 (0.0001) | 27.2 (1.25) | 1.31 (0.0001) | 33.4 (1.41) |
| DGP-ARC | 1.29 (0.05) | 1.59 (0.01) | 1.27 (0.09) | 1.65 (0.02) | 1.48 (0.03) | 1.61 (0.01) |
| DNN | 1.01 (0.001) | N/A | 1.02 (0.01) | N/A | 1.35 (0.01) | N/A |

Table A4: Test set results on the UCI benchmarks for the two hidden layer models, with standard error in brackets.

| | Powerplant | | Protein | |
|---|---|---|---|---|
| | NMSE | MNLL | NMSE | MNLL |
| DLFM | 0.0563 (0.001) | -0.0237 (0.01) | 0.706 (0.003) | 0.7335 (0.01) |
| DGP-EQ | **0.0558 (0.001)** | 0.0677 (0.06) | **0.574 (0.009)** | **0.5678 (0.03)** |
| DGP-ARC | 0.0612 (0.001) | **-0.0731 (0.01)** | 0.680 (0.030) | 0.9308 (0.01) |
| DNN | 0.0965 (0.005) | N/A | 0.614 (0.005) | N/A |

Table A5: Test set NMSE values for both FitzHugh-Nagumo experiments.

| | Scenario A - Full Data Fit | Scenario B - Extrapolation Task |
|---|---|---|
| DLFM | 0.02 | **0.07** |
| C-DGP | **0.01** | 0.38 |
| DGP-EQ | 0.03 | 0.12 |

that regardless of the scale of the initial lengthscales chosen, the two hidden layer DGPs struggled to converge during training, leading to much higher NMSE and MNLLs than those achieved by the single hidden layer DGPs in the main paper.

**UCI Regression Benchmarks** Considering the two UCI regression benchmark datasets once again, Table A4 contains the test set NMSE and MNLL for each of the four models with two hidden layers, averaged over three folds, with associated standard error reported with respect to the random seed. The inclusion of an additional hidden layer in the DLFM leads to broadly similar results to those seen for the single hidden layer case. The DGPs also tend to to follow this trend, however they converge to a considerably lower MNLL than their single hidden layer counterparts. The single hidden layer DLFM does still marginally outperform the two hidden layer DGP-EQ in terms of NMSE, and the shallow VAR-GP model from the main paper converges to a lower NMSE on the protein dataset than all of the two hidden layer models tested.

### A.3.3 FitzHugh-Nagumo System

As discussed in the main paper, the physically constrained DGP (C-DGP) of Lorenzi and Filippone [2018] is not directly applicable to problems such as the PhysioNet experiments, as the C-DGP requires that the ODE system driving the observed dynamics must be known. However, we can compare our DLFM to the C-DGP by evaluating the predictive performance of both models on a scenario in which the ODE system is known, such as the *FitzHugh-Nagumo* system [FitzHugh, 1955]. Specifically, we consider the experimental setup used by Lorenzi and Filippone [2018], which consists of 400 noisy observations of the ODE system. Firstly, we compare the performance of single hidden layer versions of both our DLFM, the C-DGP and the previously mentioned DGP-EQ on these noisy observations and evaluate the NMSE between the model predictions and the underlying ground truth, which is the exact solution of the ODE system with no noise added. These results are shown under Scenario A in Table A5. The results reported in Scenario B of Table A5 correspond to an alternate scenario in which we test the ability of all three models to extrapolate the dynamics by training on the first 300 observations and evaluating the NMSE on the final 100 observations.

From the results shown in Table A5, we can see that the C-DGP slightly outperforms the DLFM on Scenario A, likely because the C-DGP explicitly includes constraints specific to this system. However, the DLFM has a greater capacity to accurately extrapolate the dynamics despite not having any specific prior knowledge of the system, as evidenced by the results for Scenario B. The DLFM also outperforms the DGP-EQ in both tasks. The DGP-EQ tested had a single hidden layer of dimensionality 3 and 100 random features and the DLFM tested had an identical architecture but used 2 latent forces with 50 random features per latent force, for parity. The C-DGP tested had an identical architecture to that used by Lorenzi and Filippone [2018] in their experiments on this system.

Table A6: Test set NMSE values for two hidden layer models evaluated on the Lorenz attractor system.

|        | 80:20 Split | 98:2 Split |
|--------|-------------|------------|
| DLFM   | **0.94**    | **0.89**   |
| DGP-EQ | 1.09        | 1.24       |

Table A7: Test set MNLL values for the DLFM and NP models trained and evaluated on each individual output of the PhysioNet dataset in turn.

|          | ABP      | ECG      | ICP      |
|----------|----------|----------|----------|
| DLFM     | 1.53     | 1.61     | **1.51** |
| Conv-CNP | 1.59     | **1.47** | 1.84     |
| GNP      | **1.50** | 1.50     | 1.66     |

### A.3.4 Lorenz Attractor

We also conducted a brief additional experiment on a chaotic dynamical system, the *Lorenz attractor* [Lorenz, 1963],

$$\frac{dx}{dt} = \sigma(y - x)$$
$$\frac{dy}{dt} = x(r - z) - y$$
$$\frac{dz}{dt} = xy - bz$$

Utilising the commonly used system parameters $\sigma = 10$, $b = 8/3$ and $r = 28$ which result in chaotic behaviour, we generated 1000 data-points from this system in the interval $t = 0$ to $t = 50$. Two different modeling scenarios were tested, '80:20' where the first 80% of the data was used for training and the final 20% withheld for testing, and a '98:2' split where only the final 2% of the time series was withheld for testing. DLFM and DGP-EQ models identical to those used in the FitzHugh-Nagumo experiment were employed, however we utilised two hidden layers rather than one, as both models seemed to benefit from this added depth. The test set NMSE values are shown in Table A6.

We can see from these brief results presented in Table A6 that the DLFM outperforms the DGP-EQ in both scenarios, however a more rigorous series of experiments would need to be performed to accurately assess the ability of our model to model chaotic dynamical systems.

### A.3.5 Neural Process Comparison

*Neural processes* (NPs) are a recently proposed class of model, developed in order to combine a number of the benefits of neural networks and Gaussian processes [Garnelo et al., 2018]. Subsequent research in this area has focused on applications to time-series modeling by modeling *translation equivariance* in data; the *convolutional conditional neural process* (Conv-CNP) [Gordon et al., 2019] and the *Gaussian neural process* (GNP) [Bruinsma et al., 2021] are two examples of such work.

Given that time-series modeling is a key application for our proposed model, we present a brief experimental comparison between our approach and the Conv-CNP and GNP frameworks. Specifically, we considered a revised version of the PhysioNet extrapolation problem from the main paper. Following the approach taken in Section 5.1 of the work of Gordon et al. [2019], we evaluate the test set log likelihood on 1-D time series, training and evaluating a separate model for each output of the PhysioNet CHARIS dataset in turn. However, we consider a more challenging split of the training and test set data, with the first 500 data-points used for training and the final 500 withheld for testing. The deep LFM used for this experiment consisted of a single hidden layer with one latent force and 100 random Fourier features. The test set MNLL values obtained are shown in Table A7.

Table A8: Test set results averaged across all three outputs on the PhysioNet extrapolation experiment for a range of DLFM architectures. $D_F$ denotes the dimensionality of the hidden layer, $N_{RF}$ denotes the number of random Fourier features, and $Q$ denotes the number of latent forces used.

| $D_F$ | 1 | | | | 6 | | | |
|---|---|---|---|---|---|---|---|---|
| $N_{RF}$ | 10 | | 50 | | 10 | | 50 | |
| $Q$ | 1 | 6 | 1 | 6 | 1 | 6 | 1 | 6 |
| NMSE | 0.90 | 0.60 | 0.66 | 0.56 | 0.30 | 0.34 | **0.26** | 0.54 |
| MNLL | 1.52 | 1.44 | 1.41 | **1.11** | 3.34 | 3.95 | 2.02 | 1.19 |

From the results shown in Table A7, it is clear that the DLFM achieves competitive results compared to the Conv-CNP, outperforming the Conv-CNP on the ABP and ICP outputs. The GNP outperforms the DLFM on two of the outputs, but as with the DLFM comparison to the Conv-CNP, the results are relatively similar, and further analysis would be required to fully assess how the models compare under different conditions, and how their performance compares to other deep models.

### A.3.6 Hyperparameter Testing

Selecting the optimal architecture and hyperparameters for a given model is in itself a challenging research problem across sub-fields of deep learning concerned with both deterministic and probabilistic modeling. Whilst we do not provide an exhaustive assessment of the impact of certain hyperparameters on the performance of our model (primarily because this will very much depend on the data being considered), we present a brief hyperparameter study on the PhysioNet extrapolation problem from the main paper, which illustrates the impact of changing certain aspects of the DLFM. The results of this study are shown in Table A8, with the metrics averaged over all three outputs.

We can see from Table A8 that increasing the number of random Fourier features appears to reduce the predictive error and also significantly improves uncertainty quantification, as shown by the reduced MNLL values. Increasing the width of the hidden layer also broadly tends to improve performance.

### A.4 Derivation of the Variational Lower Bound

Denoting $\boldsymbol{\Psi} = \{\boldsymbol{W}, \boldsymbol{\Omega}\}$ for ease of notation, the variational lower bound on the marginal likelihood can be derived as follows:

$$
\begin{aligned}
\log[p(\boldsymbol{y}|\boldsymbol{X}, \boldsymbol{\Theta})] &= \log\left[\int p(\boldsymbol{y}|\boldsymbol{X}, \boldsymbol{\Psi}, \boldsymbol{\Theta})p(\boldsymbol{\Psi}|\boldsymbol{\Theta})d\boldsymbol{\Psi}\right] \\
&= \log\left[\int \frac{p(\boldsymbol{y}|\boldsymbol{X}, \boldsymbol{\Psi}, \boldsymbol{\Theta})p(\boldsymbol{\Psi}|\boldsymbol{\Theta})}{q(\boldsymbol{\Psi})}q(\boldsymbol{\Psi})d\boldsymbol{\Psi}\right] \\
&= \log\left[E_{q(\boldsymbol{\Psi})}\frac{p(\boldsymbol{y}|\boldsymbol{X}, \boldsymbol{\Psi}, \boldsymbol{\Theta})p(\boldsymbol{\Psi}|\boldsymbol{\Theta})}{q(\boldsymbol{\Psi})}\right] \\
&\geq E_{q(\boldsymbol{\Psi})}\left(\log\left[p(\boldsymbol{y}|\boldsymbol{X}, \boldsymbol{\Psi}, \boldsymbol{\Theta})\right]\right) + E_{q(\boldsymbol{\Psi})}\left(\log\left[\frac{p(\boldsymbol{\Psi}|\boldsymbol{\Theta})}{q(\boldsymbol{\Psi})}\right]\right) \\
&= E_{q(\boldsymbol{\Psi})}\left(\log[p(\boldsymbol{y}|\boldsymbol{X}, \boldsymbol{\Psi}, \boldsymbol{\Theta})]\right) - \mathrm{DKL}[q(\boldsymbol{\Psi})||p(\boldsymbol{\Psi}|\boldsymbol{\Theta})] \\
&\approx \left[\frac{N}{M}\sum_{k\in\mathcal{I}_M}\frac{1}{N_{\mathrm{MC}}}\sum_{r=1}^{N_{\mathrm{MC}}}\log[p(\boldsymbol{y}_k|\boldsymbol{x}_k, \tilde{\boldsymbol{\Psi}}_r, \boldsymbol{\Theta})]\right] - \mathrm{D_{KL}}[q(\boldsymbol{\Psi})||p(\boldsymbol{\Psi}|\boldsymbol{\Theta})].
\end{aligned}
$$

We assume a factorised prior over the spectral frequencies and weights across all layers, which takes the form,

$$
p(\boldsymbol{\Psi}|\boldsymbol{\Theta}) = \prod_{\ell=0}^{L-1} p\left(\boldsymbol{\Omega}^{(\ell)}|\boldsymbol{\Theta}^{(\ell)}\right) p\left(\boldsymbol{W}^{(\ell)}\right) = \prod_{ij\ell} q\left(\Omega_{ij}^{(\ell)}\right) \prod_{ij\ell} q\left(W_{ij}^{(\ell)}\right),
$$

where,

$$q\left(\Omega_{ij}^{(\ell)} \mid m_{ij}^{(\ell)}, (s^2)_{ij}^{(\ell)}\right) = \mathcal{N}\left(\Omega_{ij}^{(\ell)} \mid m_{ij}^{(\ell)}, (s^2)_{ij}^{(\ell)}\right)$$

$$q\left(W_{ij}^{(\ell)} \mid \mu_{ij}^{(\ell)}, (\beta^2)_{ij}^{(\ell)}\right) = \mathcal{N}\left(W_{ij}^{(\ell)} \mid \mu_{ij}^{(\ell)}, (\beta^2)_{ij}^{(\ell)}\right).$$

As discussed in the main paper, we reparameterize the spectral frequencies and weights as,

$$\left(\tilde{\Omega}_r^{(\ell)}\right)_{ij} = s_{ij}^{(\ell)} \epsilon_{rij}^{(\ell)} + m_{ij}^{(\ell)}$$

$$\left(\tilde{W}_r^{(\ell)}\right)_{ij} = \beta_{ij}^{(\ell)} \epsilon_{rij}^{(\ell)} + \mu_{ij}^{(\ell)}$$

where $\epsilon_{rij}^{(\ell)}$ represent samples from a standard normal, $\tilde{\Omega}_r \sim q\left(\Omega_{ij}^{(\ell)}\right)$ and $\tilde{W}_r \sim q\left(W_{ij}^{(\ell)}\right)$. We can then optimise our lower bound with respect to the parameters governing our variational distributions $\left(m_{ij}^{(\ell)}, (s^2)_{ij}^{(\ell)}, \mu_{ij}^{(\ell)} \text{ and } (\beta^2)_{ij}^{(\ell)}\right)$ and our kernel hyperparameters $\mathbf{\Theta}$, using conventional gradient descent techniques.

### A.4.1 KL Divergence Between Normal Distributions

The Kullback-Leibler (KL) divergence between two normal distributions $p_A\left(x \mid \mu_A, \sigma_A^2\right) = \mathcal{N}\left(x \mid \mu_A, \sigma_A^2\right)$ and $p_B\left(x \mid \mu_B, \sigma_B^2\right) = \mathcal{N}\left(x \mid \mu_B, \sigma_B^2\right)$ can be expressed by,

$$\mathrm{DKL}\left[p_A(x)||p_B(x)\right] = \frac{1}{2}\left[\log\left(\frac{\sigma_B^2}{\sigma_A^2}\right) - 1 + \frac{\sigma_A^2}{\sigma_B^2} + \frac{(\mu_A - \mu_B)^2}{\sigma_B^2}\right].$$

### A.5 Computational Complexity

The computational complexity associated with our stochastic variational inference scheme is $\mathcal{O}(mDQN_{RF}N_{MC})$ where $m$ denotes the mini-batch size, $D$ denotes the layer dimensionality, $Q$ and $N_{RF}$ represent the number of latent forces and random Fourier features respectively and $N_{MC}$ denotes the number of Monte Carlo samples used. This very closely follows the computational complexity of the DGP with random feature expansions presented by Cutajar et al. [2017], with an added linear dependency on $Q$, the number of latent forces employed.

### A.6 Checklist