# OpenReview forum: "Compositional Modeling of Nonlinear Dynamical Systems with ODE-based Random Features"
_NeurIPS.cc/2021/Conference — NeurIPS 2021 Poster_

### Official Review · Reviewer_Q3vp · 2021-07-12

**Rating:** 7
**Confidence:** 4

**Summary:**

This manuscript describes Deep Latent Force Models (DLFM), a nonlinear dynamical system with nicely calibrated uncertainty quantification. Similar to deep Gaussian processes (DGPs), the presented framework stacks layers of GP layers with physics-inspired covariance functions following ordinary differential equations (ODEs). The inference is based on the weight space approximation of GPs that maintains variational distributions over the weights and spectral frequencies. The model outperforms (deep) GPs on time series prediction tasks and performs slightly better in regression tasks.

**Limitations And Societal Impact:**

Both are addressed in the text.

**Main Review:**

This submission proposes a very natural extension to LFMs by "making it deeper". Although the idea and motivation are straightforward, the authors have done a great job at describing, formulating, and evaluating the method. The experiments are not very exhaustive but adequate to show clear improvements over both DGPs and LFMs. Also, the manuscript is written very clearly; I had no difficulty following the background and the methodology. All in all, the paper does not lack to address any major concerns of mine and I recommend an accept.

Below are more detailed comments and a few questions. I would be happy if these are addressed in the rebuttal (and possibly in the paper).

- The authors did not specify a recipe for choosing the form of the differential function. Since it directly impacts the covariance, gaining some intuition would be valuable. Also, is the same differential equation (line 132) used in all experiments?
-  Without any latent layer, the input to the function $f_d(t)$ was time and the ODE helped including the physics into covariance function, both of which make sense for learning dynamical systems. Now in this deep model, how can we interpret the differential equation in the latent space?
- Why is the VAR-FIXED trick needed? Also, are the same Monte Carlo samples used both for training and testing? If so, does not it imply potential overfitting to these samples?
- I wonder if the model has passed a stress test. How would deeper architectures, wider hidden layer(s), more/fewer random features, and so on affect the results, training speed/ease, etc?
- Why was LFM-RFF trained with 20 random features instead of 100 as in the deeper variant?
- Looking at Figure3b-3c and Figure5, the model seems to achieve similar MSEs as the competing methods but certainly better at uncertainty quantification. Would the authors agree with that statement? If so, stating this explicitly and a discussion on the reason(s) would help the reader better comprehend the methodology.
- In the top-left panel of Figure4, the uncertainty drops at test time (particularly when ABP<-2). Can the authors clarify why might this be happening?
- A note on the computational complexity would be nice.
- This is more of a comment than a question: Instead of this (potentially overly) complicated model, what if we had a simpler two-layer architecture that first transforms the time input non-linearly and then the final output comes from an LFM? Why should the proposed formulation be preferred over this simpler architecture (or something similar)?

**Post-rebuttal comment:** Thanks the authors for the rebuttal. I didn't have any major concerns, so I keep my rating as it is. Nice piece of work!

**Time Spent Reviewing:**

4

---

> ### Author Response · Authors · 2021-08-09
> **Response**
>
> We thank the reviewer for their time and effort in reviewing our submission and very much appreciate the positive comments regarding the methodology and manuscript itself, as well as the feedback provided. We will attempt to address each point of the feedback in turn.
>
> > **Q1** - The authors did not specify a recipe for choosing the form of the differential function. Since it directly impacts the covariance, gaining some intuition would be valuable. Also, is the same differential equation (line 132) used in all experiments?
>
> **A1** - We do indeed use the DE shown in line 132 for all of the experiments, with the main reason being that as the $\gamma$ parameter tends to infinity, it can be shown that the random features tend to those found in the deep GP with random features proposed by Cutajar et al. [1]. This is a really useful property as it means that the model can effectively automatically determine the degree to which the physics-informed features are required, and the model will revert to deep GP-like behaviour if not. In addition to this, the first order ODE is the simplest form of kernel you can select for an LFM which made it a natural starting point, although we fully intend to investigate other Green’s functions based on higher order ODEs (and even non-parametric Green’s functions) in future.
>
> > **Q2** - Without any latent layer, the input to the function fd(t) was time and the ODE helped include the physics into the covariance function, both of which make sense for learning dynamical systems. Now in this deep model, how can we interpret the differential equation in the latent space?
>
> **A2** - In a regular deep GP, the layers of the model warp the inputs, and our model does exactly the same thing, however the incorporation of the Green’s function (i.e. the LFM aspect) gives a level of flexibility to the type of warping which takes place. We can think of the Green’s function as a filter over the GPs at each layer, which selectively filters out some of the frequencies of the GPs at each layer; so, the model is not only warping the intermediate inputs but filtering components in the latent GPs making them smoother or not. When $\gamma\rightarrow\infty$, this filtering simply does not take place, as in a deep GP. This is because when $\gamma\rightarrow\infty$, the Green’s function has the effect of a Dirac delta under the integration and the random features computed are directly from the latent GP $u(t)$, which has an Exponentiated-Quadratic covariance, and the model becomes a classic deep GP model. The point made by the reviewer is a very good one in that it’s challenging to interpret these differential equations past the initial layer, but basically we are just learning the dynamics of the latent space as it allows us greater flexibility in the form of warping and filtering we perform.
>
> > **Q3** - Why is the VAR-FIXED trick needed? Also, are the same Monte Carlo samples used both for training and testing? If so, does not it imply potential overfitting to these samples?
>
> **A3** - The VAR-FIXED trick is not necessarily required, however much as Cutajar et al. [1] found, it does empirically result in better performance and improved convergence compared to the VAR-RESAMPLED approach they present (this approach specifically struggles when the number of features is increased). The Monte Carlo samples used for testing and training are not the same, so the potential overfitting mentioned is not a concern.
>
> > **Q4** - I wonder if the model has passed a stress test. How would deeper architectures, wider hidden layer(s), more/fewer random features, and so on affect the results, training speed/ease, etc?
>
> **A4** - We have extensively tested the model with many different configurations, however we felt that an experimental analysis and discussion of our findings from varying these hyperparameters and design choices might be slightly too lengthy to include in the full paper. However, we do acknowledge that sharing this information would be very useful to readers, therefore will include this within the supplemental material of the revised paper, as well as some initial results here. Typically, increasing the number of random features is a trade-off between performance and training time; we found for our experiments that 100 random features was enough to achieve impressive performance but not so many that training took an extremely long time. Additional latent forces increase the complexity of the model, which can be useful but can also lead to overfitting if increased to large values; we found that it is best to start with low values (e.g. 1 or 2) and steadily increase these if the model is not sufficiently complex. As for the number of layers, we present some results in the supplemental material which discuss the impact of adding an extra hidden layer, but will extend upon these in the revised paper, considering deeper architectures. Our initial assessment from the experiments performed was that an additional layer neither particularly helped or hindered performance, but it’s very possible more complex systems may benefit more from additional layers.
>
> As we mentioned, we will perform a more extensive analysis in the revised version of our supplemental material, as including a rigorous test over many values would generate too much data to easily present here on OpenReview. Another reviewer also requested additional information regarding hyperparameter selection, therefore we include the experimental results we gathered for that response here for the convenience of the reviewer. The results we present here were generated from different architectural choices (specifically, combinations which we didn’t include in the original paper) on the PhysioNet extrapolation problem, with the metrics averaged across all three outputs. These results are as follows:
>
> |   Width  |   1  |   1  |   1  |   1  |   6  |   6  |   6  |   6  |
> |:--------:|:----:|:----:|:----:|:----:|:----:|:----:|:----:|:----:|
> | Features |  10  |  10  |  50  |  50  |  10  |  10  |  50  |  50  |
> |  Forces  |   1  |   6  |   1  |   6  |   1  |   6  |   1  |   6  |
> |   NMSE   | 0.90 | 0.60 | 0.66 | 0.56 | 0.30 | 0.34 | 0.26 | 0.54 |
> |   MNLL   | 1.52 | 1.44 | 1.41 | 1.11 | 3.34 | 3.95 | 2.02 | 1.19 |
>
> (Note - our apologies for the table formatting, the layout would have been cleaner with merged cells but Markdown doesn’t seem to allow for this.)
>
> We can roughly see from this table that increasing the number of features tends to improve NMSE and also significantly improves the uncertainty quantification, as shown by the reduced MNLL values. Increasing the width of the hidden layer also broadly tends to improve performance.
>
> > **Q5** - Why was LFM-RFF trained with 20 random features instead of 100 as in the deeper variant?
>
> **A5** - We used 100 random features with the LFM-RFF but the results were much poorer than we anticipated, therefore we tested a number of different configurations with either one or two latent forces and either 20, 50 or 200 random features, and the model with 20 random features gave the best performance (albeit still very much inferior to the other models), so we decided to use the results from this configuration.
>
> > **Q6** - Looking at Figure3b-3c and Figure5, the model seems to achieve similar MSEs as the competing methods but certainly better at uncertainty quantification. Would the authors agree with that statement? If so, stating this explicitly and a discussion on the reason(s) would help the reader better comprehend the methodology.
>
> **A6** - Yes, we would definitely agree with that assessment and see that as one of the main strengths of the model. We will definitely state this more explicitly within the revised version of the paper to aid understanding of the methodology. As for the reasoning, we believe that the added flexibility afforded by including the $\gamma$ parameter/Green’s function (as discussed earlier in this response) allows the uncertainty to be more effectively propagated through the model.
>
> > **Q7** - In the top-left panel of Figure4, the uncertainty drops at test time (particularly when ABP<-2). Can the authors clarify why this might be happening?
>
> **A7** - It’s difficult to be sure exactly why this is happening, however we believe it may be due to the fact that in multi-output GP-based models such as ours, uncertainty intervals for certain outputs can often ‘borrow’ confidence from other outputs. For this output, although the model performs very well relative to the DGP, it does seem to make slightly conservative predictions (which can be seen from the reduced peak heights), therefore the uncertainty interval is also likely reduced because of this.
>
> > **Q8** - A note on the computational complexity would be nice.
>
> **A8** - The computational complexity is $\mathcal{O}(m N_{RF} Q p N_{MC})$, where $m$ is the minibatch size, $N_{RF}$ is the number of random features, $Q$ is the number of latent forces, $p$ is the layer dimensionality and $N_{MC}$ is the number of Monte Carlo samples used. This quite closely follows the complexity reported by Cutajar et al. [1] due to the architectural similarities, however we have an added linear dependency on the number of latent forces included.
>
> **Note - this response is continued on the following comment due to the character limit**

---

> > ### Author Response · Authors · 2021-08-09
> > **Response (continued)**
> >
> > > **Q9** - This is more of a comment than a question: Instead of this (potentially overly) complicated model, what if we had a simpler two-layer architecture that first transforms the time input non-linearly and then the final output comes from an LFM? Why should the proposed formulation be preferred over this simpler architecture (or something similar)?
> >
> > **A9** - This is an interesting idea and somewhat resembles a ‘deep kernel learning’ approach but with an LFM in place of the GP, it’s an approach that has crossed our mind during this project. There’s no guarantee that one would be better than the other but we believe that this approach may be preferable due to the reasons given previously, that the flexibility afforded by forming compositions of these Green’s function ‘filters’ allows the latent dynamics to be learned in an effective manner. Whilst this is an area we plan to perform further research in, we did perform a brief experiment to investigate this idea, forming a DLFM with two hidden layers, but with the features at the first hidden layer replaced with the DGP random features used in [1] (denoted ‘Deep GP-LFM’). This is not exactly the scenario you’ve posed but it’s similar, as the layer of DGP features is acting as a non-linearity, the output of which is passed to a layer of LFM-based features. We also consider for completeness, the reverse formulation, with an initial layer of LFM features, followed by a layer of DGP features, denoted ‘Deep LFM-GP’. We compare both of these models to the two hidden layer DLFM results presented in the appendix of our paper for the PhysioNet extrapolation problem. We consider DLFMs with Q=1, with all other aspects/hyperparameters of the DLFM are kept the same, and the DGP features used have an exponentiated quadratic (EQ) kernel. We omit comparison to the DGP from [1], as the two layer version of this model struggled to converge when tested on this problem.
> >
> > |             |      |  ABP |  ECG |  IBP |
> > |:-----------:|:----:|:----:|:----:|:----:|
> > | Deep GP-LFM | NMSE | 0.43 | 0.40 | 0.50 |
> > | Deep GP-LFM | MNLL | 0.78 | 0.96 | 1.19 |
> > | Deep LFM-GP | NMSE | 1.09 | 1.04 | 1.06 |
> > | Deep LFM-GP | MNLL | 1.47 | 1.48 | 1.44 |
> > |     DLFM    | NMSE | 0.26 | 0.55 | 0.35 |
> > |     DLFM    | MNLL | 0.45 | 1.03 | 0.92 |
> >
> > We can see from these results that whilst the DLFM outperforms the Deep GP-LFM on two of the three outputs (possibly as a result of the reason suggested above), the Deep GP-LFM still performs very competitively, especially compared to the Deep LFM-GP model. As mentioned, we plan to explore these ideas in considerably more depth in future work, and from these initial results, it seems this research would be a worthwhile endeavour. We would happily include the above analysis in the revised appendix of our paper if the reviewer thinks this would be appropriate.
> >
> > Again, we thank the reviewer for their time and hope our comments have addressed your feedback. If you have any further questions regarding the submission then please let us know.
> >
> > [1] Cutajar, K., Bonilla, E.V., Michiardi, P. and Filippone, M., 2017, July. Random feature expansions for deep Gaussian processes. In International Conference on Machine Learning (pp. 884-893). PMLR.

---

### Official Review · Reviewer_L4Ej · 2021-07-15

**Rating:** 7
**Confidence:** 3

**Summary:**

This paper introduces a model aimed to learn time series stemming from Dynamical Systems while incorporating uncertainty. The model is a Deep Gaussian Process where the random features expansion is done using a physics informed approximation derived from the Green Kernels of certain families of ODEs.

**Limitations And Societal Impact:**

I don't think there are any other than what is already acknowledged in the paper.

**Main Review:**

The model presented here is novel and its derivation seems principled enough in my opinion. The results show that DLFM compares favourably to DGPs both in terms of accuracy and uncertainty calibration.

However, I am quite skeptical about the baselines used here: how is a NN with a single hidden layer deep? There are many other architectures (deeper, residual, recurrent, Neural ODE-like models,...) which can be compared in terms of accuracy and sometimes uncertainty calibration to the proposed model. More specifically, I think that the model presented in https://openreview.net/forum?id=Skey4eBYPS is particularly relevant to this work and should be presented as a baseline.

Thus, while I think the ideas proposed here are promising indeed, I don't think that the baselines compared to are strong enough to fully validate them.

Edit after rebuttal: After reading other reviews and the answer of the authors, I am happy to raise my score: This paper brings a novel method to GP-based estimation of non-linear DS which should be of interest to many in the community.

**Time Spent Reviewing:**

3

---

> ### Author Response · Authors · 2021-08-09
> **Response**
>
> We thank the reviewer for their time and effort reviewing our submission and for the positive comments on the model itself, as well as the feedback regarding baselines which we will attempt to address here.
>
> > **Q1** - However, I am quite skeptical about the baselines used here: how is a NN with a single hidden layer deep? There are many other architectures (deeper, residual, recurrent, Neural ODE-like models,...) which can be compared in terms of accuracy and sometimes uncertainty calibration to the proposed model. More specifically, I think that the model presented in https://openreview.net/forum?id=Skey4eBYPS is particularly relevant to this work and should be presented as a baseline. Thus, while I think the ideas proposed here are indeed promising, I don't think that the baselines compared to are strong enough to fully validate them.
>
> **A1** - We decided to initially experiment with deep LFMs, deep GPs and DNNs consisting of a single hidden layer in order to assess the performance of the simplest possible architecture for each model type, before moving on to consider more complex versions of said models with an additional hidden layer; our results for the two hidden layers models are included in our appendix. This is an approach also taken by other works in the deep GP literature such as Cutajar et al. [1], Salimbeni & Deisenroth [2] and Lorenzi & Filippone [3] (the latter actually only considers a single hidden layer). However, we do appreciate that including experimental results for deeper architectures would be useful information to include for the reader, and will include an expanded analysis of deeper architectures in the revised version of our paper.
>
> We chose to compare mainly against deep GP, GP and LFM baselines as one of the main aims of our work was to improve the ability of GP-based methods to accurately model and extrapolate complex dynamics, so making comparison to these other methods was particularly important. However, we do acknowledge that consideration of other deep, non-GP-based baselines would give a broader sense of how the model compares to other recent advances in modeling dynamical systems. We intend to include a more thorough experimental analysis with the types of model suggested by the reviewer in the revised version of our paper, with a particular focus on the convolutional conditional neural process as suggested. We include here a brief preliminary analysis of some of these suggested baselines on a revised version of the PhysioNet extrapolation problem from our paper. Following the approach taken in section 5.1 of the convolutional conditional neural processes paper, of evaluating the test set log likelihood on 1-D time series, we train and evaluate a separate model for each output in turn. However, we consider a more challenging 50:50 train:test set split, with the first 500 data-points used for training and the final 500 for testing. The test set MNLL values were as follows:
>
> |          |  ABP |  ECG |  IBP |
> |:--------:|:----:|:----:|:----:|
> |   DLFM   | 1.53 | 1.61 | 1.51 |
> | Conv-CNP | 1.59 | 1.47 | 1.84 |
> |    GNP   | 1.50 | 1.50 | 1.66 |
>
> ‘DLFM’ here is a single hidden layer deep LFM with one latent force and 100 random features and ‘Conv-CNP’ is a convolutional conditional neural process as implemented by Gordon & Bruinsma et al. [4]. We also consider the Gaussian neural process (‘GNP’) proposed by Bruinsma et al. [5], which is a more recent model which builds upon the work of [4]. From these initial results, it appears that the DLFM achieves competitive results compared to the Conv-CNP, outperforming the Conv-CNP on the ABP and IBP outputs. The GNP outperforms the DLFM on two of the outputs, but as with the DLFM comparison to the Conv-CNP, the results are relatively similar, and further analysis will be required to fully assess how the models compare under different conditions, and how they stack up against other deep models (which we will include in our revised paper, as we mentioned). We do hope however that this initial comparison to two very effective recent models provides some further validation for the effectiveness of our approach.
>
> Once again, we thank the reviewer for their time and hope our comments have addressed your concerns. We are happy to answer any further questions you may have about the submission.
>
> [1] Cutajar, K., Bonilla, E.V., Michiardi, P. and Filippone, M., 2017, July. Random feature expansions for deep Gaussian processes. In International Conference on Machine Learning (pp. 884-893). PMLR.
>
> [2] Salimbeni, H. and Deisenroth, M., 2017. Doubly stochastic variational inference for deep Gaussian processes. arXiv preprint arXiv:1705.08933.
>
> [3] Lorenzi, M. and Filippone, M., 2018, July. Constraining the dynamics of deep probabilistic models. In International Conference on Machine Learning (pp. 3227-3236). PMLR.
>
> [4] Gordon, J., Bruinsma, W.P., Foong, A.Y., Requeima, J., Dubois, Y. and Turner, R.E., 2019. Convolutional conditional neural processes. arXiv preprint arXiv:1910.13556.
>
> [5] Bruinsma, W.P., Requeima, J., Foong, A.Y., Gordon, J. and Turner, R.E., 2021. The Gaussian Neural Process. arXiv preprint arXiv:2101.03606.

---

### Official Review · Reviewer_YdjF · 2021-07-16

**Rating:** 6
**Confidence:** 3

**Summary:**

This paper introduces a deep GP for nonlinear dynamical systems. Crucially, each of the GPs uses a physics-informed kernel a la the latent force model.

**Limitations And Societal Impact:**

The authors adequately addressed the limitations and potential negative societal impact of their work.

**Main Review:**

# Strengths
Learning good models of nonlinear dynamical systems is an important topic that spans many fields such as RL and state-space models to name a few. I think this is a novel combination of deep GPs + random features + latent force models that allows for fast and efficient learning and inference.

# Weaknesses
I think the biggest weakness is the structure of the paper. Firstly, I don't think the idea of using physics-informed random features was well motivated. Specifically, for a simple latent force model the functional form of the dynamics is assumed--providing a very strong inductive bias--and a GP is used to learn the force components. Or for deep CNN, the pro of using multiple convolutional layers is that they extract a hierarchy of visual features. It would have been nice if the authors would have demonstrated some empirical evidence that these physics-informed random features give the model an inductive bias towards certain types of dynamics.

Next, I think there was unnecessary notation that made the paper harder to follow. Specifically, in section 2.2 the authors introduce random feature expansions for latent force models with a generic Green's functions but in section 3.1 they end up using the Green's function for a linear ODE; this led to an increase in, my opinion, unnecessary notation that just made the approach harder to follow.

Lastly, I am puzzled by the choice of the UCI regression benchmarks task in the experiments section. The goal of this paper was to introduce a method for learning dynamical systems. While I am impressed that the proposed approach works well here, these types of datasets isn't what the approach was motivated for (which ties back to my point about lack of motivation). For instance, there are a number of synthetic dynamical systems that are commonly used to benchmark sequential VAEs (these can be adapted to this instance by having the latent dynamics be fully observed): FitzHugh-Nagumo, Lorenz attractor, Cart-pole, different robots from MuJoCo, etc. I would have been more impressed seeing these types of results in a paper about learning nonlinear dynamical systems.

# Conclusion
While I do think this is a cool idea, I think more time needs to be spent on the writing of the paper and applying the proposed approach on common nonlinear dynamical systems.


**Time Spent Reviewing:**

6 hours

---

> ### Author Response · Authors · 2021-08-09
> **Response**
>
> We thank the reviewer for their time and effort in reviewing our paper, especially for their useful feedback, which we will attempt to address here.
>
> > **Q1** - I think the biggest weakness is the structure of the paper. Firstly, I don't think the idea of using physics-informed random features was well motivated. Specifically, for a simple latent force model the functional form of the dynamics is assumed--providing a very strong inductive bias--and a GP is used to learn the force components. Or for deep CNN, the pro of using multiple convolutional layers is that they extract a hierarchy of visual features. It would have been nice if the authors would have demonstrated some empirical evidence that these physics-informed random features give the model an inductive bias towards certain types of dynamics.
>
> **A1** - Similarly to deep GPs, the layers within our model warp the input space, propagating uncertainty through the model. The motivation for selecting the first order ODE form for the random features is that as the $\gamma$ parameter in the ODE tends to infinity, it can be shown that the random features tend to those found in the deep GP with random features proposed by Cutajar et al. [1]. Indeed, when $\gamma\rightarrow\infty$, the Green’s function has the effect of a Dirac delta under the integration and the random features computed are directly from the latent GP $u(t)$, which has an Exponentiated-Quadratic covariance, and the model becomes a classic deep GP model. This is key as it means that the model effectively can automatically select the degree to which the inductive bias introduced by the physics-informed random features is required. Alternatively, the Green’s function can also be interpreted as a filter, which selectively filters out some of the frequencies of the GPs at each layer; so, the model is not only warping the intermediate inputs but filtering components in the latent GPs making them smoother or not. When $\gamma\rightarrow\infty$, this filtering simply does not take place, as in a deep GP. An even more flexible scenario would be for us to have a non-parametric form of the Green’s function, and this is something that we are currently investigating as further work based on this paper.
>
> > **Q2** - Next, I think there was unnecessary notation that made the paper harder to follow. Specifically, in section 2.2 the authors introduce random feature expansions for latent force models with a generic Green's functions but in section 3.1 they end up using the Green's function for a linear ODE; this led to an increase in, my opinion, unnecessary notation that just made the approach harder to follow.
>
> **A2** - We politely disagree with the reviewer. Indeed, we could have taken this approach, starting from the first order ODE, but we would have lost the generality of the approach which makes it clear that this methodology can be employed with alternative forms of the Green’s function, such as those corresponding to higher order or partial differential equations, or even a non-parametric form of the Green’s function. We already have code for a second order ODE version of this model and are currently developing a version of the model with a non-parametric Green’s function, however as this is an initial work and we did not have space to discuss these approaches in depth, we have focused solely on the first order ODE model in this work. We do however fully acknowledge that formulating the notation in this manner does make the work more challenging to understand, and we will simplify the notation in the manner suggested by the reviewer in the final version of the paper.
>
> > **Q3** - Lastly, I am puzzled by the choice of the UCI regression benchmarks task in the experiments section. The goal of this paper was to introduce a method for learning dynamical systems. While I am impressed that the proposed approach works well here, these types of datasets isn't what the approach was motivated for (which ties back to my point about lack of motivation). For instance, there are a number of synthetic dynamical systems that are commonly used to benchmark sequential VAEs (these can be adapted to this instance by having the latent dynamics be fully observed): FitzHugh-Nagumo, Lorenz attractor, Cart-pole, different robots from MuJoCo, etc. I would have been more impressed seeing these types of results in a paper about learning nonlinear dynamical systems.
>
> **A3** - We chose to include the UCI benchmarks to illustrate the point mentioned earlier in this comment, that when the physics-informed features are not required, the model reverts to deep GP-like behaviour and still performs well; we were also impressed by the results which also factored into our decision to include them as they were particularly interesting. However, we do fully acknowledge the fact that benchmarking on other nonlinear dynamical systems is well worth investigating, and to that end we have benchmarked the model on some of the datasets suggested, and will include these results within the revised version of the paper.
>
> Another reviewer commented on a comparison for which we ran out an experiment using the FitzHugh-Nagumo system, therefore we include those results here for the convenience of the reviewer. Specifically we use the FitzHugh-Nagumo data provided in the [Github repo for the paper](https://github.com/mauriziofilippone/constraining_dynamics_deep_models/tree/master/ICML2018_code), comparing the performance of our model to that of Lorenzi & Filippone [2] (denoted ‘C-DGP’), and also to the deep GP of Cutajar et al [1] (denoted ‘RFF-DGP’). We will expand upon this brief analysis in the revised version of the paper. Specifically, we use the data provided at the link above for the ‘0’ experiment, which consists of 400 noisy observations of the ODE system; predictive means and standard deviations for a single layer constrained DGP are also provided at the link. As in [2], we firstly train single hidden layer versions of our model and that of [1] on these noisy observations and evaluate the NMSE between the model predictions and the underlying ground truth (i.e. the exact solution of the ODE system with no noise added); these results are under column ‘A’. In column ‘B’, we test the ability of all three models to extrapolate the dynamics by training on the first 300 observations and evaluating the NMSE on the final 100.
>
> |         | A - Full Data Fit | B- Extrapolation Task |
> |:-------:|:-----------------:|:---------------------:|
> |   DLFM  |        0.02       |          0.07         |
> |  C-DGP  |        0.01       |          0.38         |
> | RFF-DGP |        0.03       |          0.12         |
>
> We see here that the C-DGP slightly outperforms our DLFM in the ‘A’ scenario (as we might expect given that they explicitly include constraints specific to this system, whereas we do not), however our model has a much greater capacity to accurately extrapolate the dynamics, as shown in the results for task ‘B’. We can also see here that our model outperforms the RFF-DGP in both scenarios. The RFF-DGP tested had a single hidden layer of dimensionality 3 and 100 random features; the DLFM tested had an identical architecture but used 2 latent forces with 50 random features per latent force (100 in total, for parity).
>
> Secondly, we performed some experiments on the Lorenz attractor system as suggested; 1000 data-points were generated from the system in the interval $t=0$ to $t=50$, with the commonly used system parameters $a=28$, $r=10$ and $b=8/3$. We tested two scenarios, ‘80:20’ where the first 80% of the data was used for training and the final 20% withheld for testing, and a ‘98:2’ split where only the final 2% of the time series was withheld for testing. DLFM and RFF-DGP models identical to that used in the FitzHugh-Nagumo experiment were employed, however we utilised two hidden layers rather than one, as both models seemed to benefit from this added depth. The test set NMSE values were as follows:
>
> |         | 80:20 Split | 98:2 Split |
> |---------|:-----------:|:----------:|
> | DLFM    |     0.94    |    0.89    |
> | RFF-DGP |     1.09    |    1.24    |
>
> We can see here that the DLFM outperforms the RFF-DGP in both cases. We were unable to evaluate the C-DGP on this data as doing so would require system-specific constraints to be derived and integrated into the code for their model.
>
> We once again thank the reviewer for their time and hope our comments have addressed your concerns regarding the submission. We are happy to answer any further questions you may have.
>
> [1] Cutajar, K., Bonilla, E.V., Michiardi, P. and Filippone, M., 2017, July. Random feature expansions for deep Gaussian processes. In International Conference on Machine Learning (pp. 884-893). PMLR.
>
> [2] Lorenzi, M. and Filippone, M., 2018, July. Constraining the dynamics of deep probabilistic models. In International Conference on Machine Learning (pp. 3227-3236). PMLR.

---

> > ### Comment · Reviewer_YdjF · 2021-08-30
> > **Response to authors**
> >
> > Thank you to the authors for meticulously responding to my review. I appreciate the authors running the proposed model on FitzHugh-Nagumo and the Lorenz attarctor, which I think substantially beefs up the experimental section. I think the paper would benefit from a paragraph/subsection better elucidating the points made in A1, as it personally helped me.
> >
> > I will raise my score accordingly.

---

> > > ### Author Response · Authors · 2021-09-01
> > > **Response to reviewer comment**
> > >
> > > We thank the reviewer for their additional feedback and are glad that the points made in A1 were particularly useful for the reviewer; we will certainly include the suggested paragraph expanding upon these points in the revised version of our paper.

---

### Official Review · Reviewer_aNaL · 2021-07-19

**Rating:** 6
**Confidence:** 2

**Summary:**

This paper introduces a method for incorporating physical dynamics into Deep GPs for the purpose of modeling nonlinear dynamical systems. The dynamics are incorporated through a kernel built via the Green's function for an associated dynamical system.

**Main Review:**

Overall, the paper was well motivated and well written.  My main concerns are have to do with related work:

First, I would have appreciated additional summary of how this method compares to previous work. The introduction and related work section discuss a number of methods and techniques that this work builds off of or is similar to (LFMs, DGPs, random Fourier features, etc) as well as different axes along which the related works differ (shallow vs deep, if or how they incorporate physical dynamics, etc). Perhaps include a table that enumerates these related works and which combination(s) of the techniques they studied?

Second, the experimental section could make additional comparisons to other methods for modeling nonlinear dynamical systems. The paper mentions the work of Lorenzi and Filippone as being quite similar, but having a different approach to incorporating dynamical constraints. Shouldn't their method also be included in the experimental evaluation?

Finally, the paper could provide more guidance on how to choose the number of latent forces and the number of random features, in addition to other hyperparameters. How do these choices affect the results?

**Time Spent Reviewing:**

5

---

> ### Author Response · Authors · 2021-08-09
> **Response**
>
> We very much appreciate the time and effort taken to review our work and appreciate the positive comments, as well as the useful feedback.
>
> > **Q1** - First, I would have appreciated an additional summary of how this method compares to previous work. The introduction and related work section discuss a number of methods and techniques that this work builds off of or is similar to (LFMs, DGPs, random Fourier features, etc) as well as different axes along which the related works differ (shallow vs deep, if or how they incorporate physical dynamics, etc). Perhaps include a table that enumerates these related works and which combination(s) of the techniques they studied?
>
> **A1** - We do appreciate the fact that as we cover quite a number of comparisons to other methods such as Cutajar et al. [1], Mehrkanoon and Suykens [2] and Lorenzi and Filippone [3] in the introduction along with Wang et al. [4], Zammit-Mangion et al. [5] and Duncker and Sahani [6] in the related work section, such a table would be an intuitive and useful way for readers to quickly gain an appreciation at a glance for how our model compares to other techniques. We welcome this suggestion and will include such a table in the final version of the paper listing all techniques discussed, noting whether they are deep/shallow, probabilistic/deterministic, if they incorporate physical dynamics and finally a brief description of how any physical dynamics are incorporated.
>
> > **Q2** - Second, the experimental section could make additional comparisons to other methods for modeling nonlinear dynamical systems. The paper mentions the work of Lorenzi and Filippone as being quite similar, but having a different approach to incorporating dynamical constraints. Shouldn't their method also be included in the experimental evaluation?
>
> **A2** - We mention (albeit in the supplemental material) that we chose not to compare to Lorenzi and Fillippone [3] as their modeling approach involves performing parameter inference given some **known** restrictions/constraints based on a **known** system of ODEs. However, in our work, we do not assume knowledge of the ODE system driving the dynamics, therefore we do not attempt to characterise the system fully and perform parameter inference. We are more concerned with creating a model which can make accurate predictions and extrapolate nonlinear dynamics effectively in scenarios in which characterising all of the interactions via an ODE system is not feasible. However, we do acknowledge that a comparison to [3] would still be a beneficial addition to our work, therefore we performed a comparison to this work on the FitzHugh-Nagumo data provided in the [Github repo for the paper](https://github.com/mauriziofilippone/constraining_dynamics_deep_models/tree/master/ICML2018_code), comparing the performance of our model to that of Lorenzi & Filippone [3] (denoted ‘C-DGP’), and also to the deep GP of Cutajar et al. [1] (denoted ‘RFF-DGP’). We briefly summarize the results next, but we will include an expanded version of this analysis in our revised paper.
> Specifically, we use the data provided at the link above for the ‘0’ experiment, which consists of 400 noisy observations of the ODE system; predictive means and standard deviations for a single layer constrained DGP are also provided at the link. As in [3], we firstly train single hidden layer versions of our model and that of [1] on these noisy observations and evaluate the NMSE between the model predictions and the underlying ground truth (i.e. the exact solution of the ODE system with no noise added); these results are under column ‘A’. In column ‘B’, we test the ability of all three models to extrapolate the dynamics by training on the first 300 observations and evaluating the NMSE on the final 100.
>
> |         | A - Full Data Fit | B- Extrapolation Task |
> |:-------:|:-----------------:|:---------------------:|
> |   DLFM  |        0.02       |          **0.07**         |
> |  C-DGP  |        **0.01**       |          0.38         |
> | RFF-DGP |        0.03       |          0.12         |
>
> We see here that the C-DGP slightly outperforms our DLFM in the ‘A’ scenario (as we might expect given that they explicitly include constraints specific to this system, whereas we do not), however our model has a greater capacity to accurately extrapolate the dynamics, as shown in the results for task ‘B’. We can also see here that our model outperforms the RFF-DGP in both scenarios. The RFF-DGP tested had a single hidden layer of dimensionality 3 and 100 random features; the DLFM tested had an identical architecture but used 2 latent forces with 50 random features per latent force (100 in total, for parity).
> As a byproduct of constraining the dynamics of their model, Lorenzi & Filippone end up using an alternative set of random features that often consist of taking the derivative of the original features used in the ‘RFF-DGP’. Following this idea, we performed an additional version of the PhysioNet extrapolation experiment from our paper using a version of our model where the random features were replaced with their derivatives (otherwise, the model is identical to that used for the same experiment in the paper) in order to investigate the impact of this design choice on performance. The results were as follows:
>
> |     | Original DLFM | Original DLFM | Derivative DLFM | Derivative DLFM |
> |:---:|:-------------:|:-------------:|:---------------:|:---------------:|
> |     |      NMSE     |      MNLL     |       NMSE      |       MNLL      |
> | ABP |      **0.39**     |      **0.75**     |       0.79      |       1.08      |
> | ECG |      **0.45**     |      **0.99**     |       0.92      |       1.38      |
> | IBP |      **0.36**     |      **0.90**     |       0.50      |       1.06      |
>
> (Note - our apologies for the table formatting, the layout would have been cleaner with merged cells but Markdown doesn’t seem to allow for this.)
>
> From this we see that solely using the derivative features does not improve performance over the original DLFM. Obviously we aren’t applying explicit constraints here a la Lorenzi & Filippone (as we can’t, since we have no knowledge of the underlying system), but we thought this was an interesting comparison to make, and we will expand on this in the appendix of our revised paper.
>
> > **Q3** - Finally, the paper could provide more guidance on how to choose the number of latent forces and the number of random features, in addition to other hyperparameters. How do these choices affect the results?
>
> **A3** - As with many research areas focused on deep learning and deep probabilistic models, selecting the optimal architecture and hyperparameters is a challenging problem, and often the optimal choices depend heavily on the dataset used. For this reason, we plan to do further work on improving the model selection process for these deep LFMs. However, we agree that including further discussion of these hyperparameters would be useful to readers and will include this in the revised paper or supplemental material. To briefly summarise, selection of the number of random features is effectively a trade-off between training time and performance; adding more features tends to improve performance at the expense of increased training time, and for the datasets we used in the paper we found 100 random features to be a good compromise between the two. Additional latent forces increase the complexity of the model, which can be useful but can also lead to overfitting if increased to large values; we found that it is best to start with low values (e.g. 1 or 2) and steadily increase these if the model is not sufficiently complex. As for the number of layers, we present some results in the supplemental material which discuss the impact of adding an extra hidden layer. We found in our experiments that it neither particularly helped or hindered performance but it’s possible more complex systems may benefit more from additional layers.
> As we mentioned, we will perform a more extensive analysis in the revised version of our paper or supplemental material, as including a rigorous test over many values would generate too much data to easily present here on OpenReview. However, we present some results here generated from different architectural choices (specifically, combinations which we didn’t include in the original paper) on the PhysioNet extrapolation problem, with the metrics averaged across all three outputs. These results are as follows:
>
> |   Width  |   1  |   1  |   1  |   1  |   6  |   6  |   6  |   6  |
> |:--------:|:----:|:----:|:----:|:----:|:----:|:----:|:----:|:----:|
> | Features |  10  |  10  |  50  |  50  |  10  |  10  |  50  |  50  |
> |  Forces  |   1  |   6  |   1  |   6  |   1  |   6  |   1  |   6  |
> |   NMSE   | 0.90 | 0.60 | 0.66 | 0.56 | 0.30 | 0.34 | **0.26** | 0.54 |
> |   MNLL   | 1.52 | 1.44 | 1.41 | **1.11** | 3.34 | 3.95 | 2.02 | 1.19 |
>
> We can roughly see from this table that increasing the number of features tends to improve NMSE and also significantly improves the uncertainty quantification, as shown by the reduced MNLL values. Increasing the width of the hidden layer also broadly tends to improve performance.
>
> Once again, we thank the reviewer for their time and hope our comments have addressed your points of feedback. If you have any further questions about the submission then we are happy to answer them.
>
> **Note - References are provided in the following comment due to character limit**

---

> > ### Author Response · Authors · 2021-08-09
> > **Response (continued, to add references)**
> >
> > [1] Cutajar, K., Bonilla, E.V., Michiardi, P. and Filippone, M., 2017, July. Random feature expansions for deep Gaussian processes. In International Conference on Machine Learning (pp. 884-893). PMLR.
> >
> > [2] Mehrkanoon, S. and Suykens, J.A., 2018. Deep hybrid neural-kernel networks using random Fourier features. Neurocomputing, 298, pp.46-54.
> >
> > [3] Lorenzi, M. and Filippone, M., 2018, July. Constraining the dynamics of deep probabilistic models. In International Conference on Machine Learning (pp. 3227-3236). PMLR.
> >
> > [4] Wang, Y., Smola, A., Maddix, D., Gasthaus, J., Foster, D. and Januschowski, T., 2019, May. Deep factors for forecasting. In International conference on machine learning (pp. 6607-6617). PMLR.
> >
> > [5] Zammit-Mangion, A., Ng, T.L.J., Vu, Q. and Filippone, M., 2021. Deep compositional spatial models. Journal of the American Statistical Association, pp.1-22.
> >
> > [6] Duncker, L. and Sahani, M., 2018. Temporal alignment and latent Gaussian process factor inference in population spike trains. bioRxiv, p.331751.

---

> > > ### Comment · Reviewer_aNaL · 2021-08-30
> > > **Thank you for your response**
> > >
> > > Thank you for your response, the comparisons on a system with known dynamics (the FiztHugh-Nagumo model) is a nice addition.

---

### Decision · Program_Chairs · 2021-09-27

**Decision:**

Accept (Poster)

**Comment:**

There was initially some spread in the reviewer scores and perception of this paper, but the reviewer consensus did lean towards acceptance in the end. The additions proposed in the rebuttal are quite extensive but written out in sufficient detail that the reviewers also believe the paper can be accepted. Please, carefully pay attention to the reviewer comments and make sure to include the promised changes in the camera-ready version. In terms of additional experiments, including the results you presented during the rebuttal/discussion phase is sufficient.